

# Anthropocene Climate Bifurcation

Kolja Leon Kypke, William Finlay Langford, and Allan Richard Willms

Department of Mathematics and Statistics, University of Guelph, 50 Stone Road West, Guelph, ON, Canada N1G 2W1

**Correspondence:** William F. Langford (wlangfor@uoguelph.ca)

**Abstract.** This article presents the results of a bifurcation analysis of a simple Energy Balance Model (EBM) for the future climate of the Earth. The main focus is on the question: Can the nonlinear processes intrinsic to atmospheric physics, including natural positive feedback mechanisms, cause a mathematical bifurcation of the climate state, as a consequence of continued anthropogenic forcing by rising greenhouse gas emissions? Our analysis shows that such a bifurcation could cause an abrupt

change, to a drastically different climate state in the EBM, warmer and more equable than any climate existing on Earth since the Pliocene Epoch. In previous papers, with this EBM adapted to paleoclimate conditions, it was shown to exhibit saddlenode and cusp bifurcations. The EBM was validated by the agreement of its predicted bifurcations with the abrupt climate changes that are known to have occurred in the paleoclimate record, in the Antarctic at the Eocene-Oligocene Transition (EOT) and in the Arctic at the Pliocene-Paleocene Transition (PPT). In this paper, the EBM is adapted to fit Anthropocene climate conditions,

with emphasis on the Arctic and Antarctic climates. The four Representative Concentration Pathways (RCP) developed by the IPCC are used to model future $CO_2$ concentrations, corresponding to different scenarios of anthropogenic activity. In addition, the EBM investigates four naturally occurring nonlinear feedback processes which magnify the warming that would be caused by anthropogenic $CO_2$ emissions alone. These four feedback mechanisms are: ice-albedo feedback, water vapour feedback, ocean heat transport feedback and atmospheric heat transport feedback. The EBM predicts that a bifurcation resulting in

a catastrophic climate change will occur in coming centuries, for an RPC with unabated anthropogenic forcing, amplified by these positive feedbacks. However, the EBM suggests that appropriate reductions in carbon emissions may limit climate change to a more tolerable continuation of what is observed today. This EBM has an Equilibrium Climate Sensitivity (ECS) near the high end of the range recorded by the IPCC.

## 1  Introduction

Today, there is widespread agreement that the climate of the Earth is changing, but the precise trajectory of future climate change is still a matter of debate. Recently there has been much interest in the possibility of tipping points (or bifurcation points) causing abrupt changes in the evolution of the Earth climate system, see (Alley et al. (2003); Seager and Battisti (2007); Lenton et al. (2008); Ditlevsen and Johnsen (2010); Lenton (2012); Barnosky et al. (2012); Dijkstra (2013); Drijfhout





et al. (2015); Bathiany et al. (2016); North and Kim (2017); Steffen et al. (2018); Wallace-Wells (2019)). Section 12.5.5 in (IPCC (2013)) gives an overview of such potential abrupt changes. At such points, a small change in the forcing parameters (whether anthropogenic or natural forcings) may cause a catastrophic change in the state of the system. In order to prepare for future climate change, it is of great importance to know if such abrupt changes can occur, and if so, when and how they will occur. It has been suggested that conventional General Circulation Models (GCM) may be "too stable" to provide reliable

warning of these sudden catastrophic events (Valdes (2011)), but the study of paleoclimates may be a better guide to how abrupt changes may occur (Zeebe (2011)). In this paper, a simple Energy Balance Model (EBM) is used to investigate the possibility of such sudden catastrophic events.

In the literature, EBMs have played an important role in understanding climate and climate change (Budyko (1968); Sellers (1969); Sagan and Mullen (1972); North et al. (1981); Thorndike (2012); Kaper and Engler (2013); Dijkstra (2013); Payne et

al. (2015); Hartmann (2016); North and Kim (2017)). Often these EBMs do exhibit bifurcations, but may be lacking in the geophysical details and the mathematical rigour required to make useful predictions. This paper presents an EBM built upon basic laws of geophysics, which determine nonlinear positive feedback processes that amplify anthropogenic $CO_2$ forcing and may lead to bifurcations. A mathematical bifurcation analysis of this EBM applied to paleoclimate changes was presented in (Kypke (2019); Kypke and Langford (2020)). That analysis gave a mathematical proof of the existence of a cusp bifurcation

in the EBM. The existence of the cusp bifurcation implies the co-existence of two distinct stable equilibrium climate states (bistability), as well as the existence of abrupt transitions between these two states (via fold bifurcations) in the EBM. These transitions are dependent on the past history of the system (hysteresis). In addition, the two universal unfolding parameters for the cusp bifurcation were determined as functions of the relevant physical parameters.

One advantage of an EBM over a more complex GCM is that it facilitates the exploration of specific cause and effect relation-

ships, as particular climate forcing factors are varied or ignored. Another advantage of an EBM is that rigorous mathematical analysis often can prove the existence of certain behaviours, such as bistability and bifurcations, that could only be surmised from numerical evidence, or missed, in more complicated models. Four versions of the EBM are considered here: a globally averaged temperature model and three regional models corresponding to Arctic, Antarctic and Tropical climates.

This EBM was validated in (Dortmans et al. (2019); Kypke and Langford (2020)), where it was applied to known paleocli-

mate changes. It successfully "predicted" the abrupt glaciation of Antarctica at the Eocene-Oligocene transition and the abrupt glaciation of the Arctic at the Pliocene-Pleistocene transition, using a bifurcation analysis. Those paleoclimate bifurcations led to abrupt cooling, from warm, equable "hothouse" climate conditions to a cooler state like the climate of today, with ice-capped poles, as recorded in the geological record.

This paper applies that paleoclimate EBM to the Earth's climate of the Anthropocene. It explores the possibility of a "re-

versal" of those two paleoclimate bifurcations, which may (or may not) occur in future centuries, from today's climate to an equable hothouse climate state, such as existed in the Pliocene and earlier. It provides new mathematical evidence signifying that catastrophic climate change in polar regions is inevitable in the coming decades and centuries, if current anthropogenic forcing continues unabated. The EBM also suggests that mitigation strategies exist, which can avoid such an outcome.


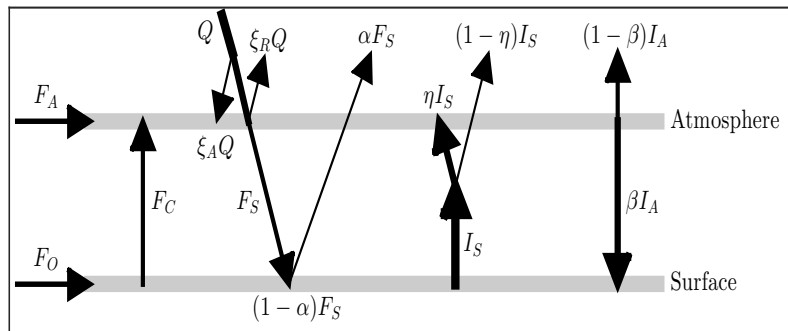

**Figure 1.** A visualization of the energy balance model (EBM). Here $Q$ is the incident solar radiation. A fraction $\xi_A$ of $Q$ is absorbed by the atmosphere and another fraction $\xi_R$ is reflected by clouds into space. The resulting solar forcing that strikes the surface is $F_S = (1 - \xi_A - \xi_R)Q$. The surface has albedo $\alpha$, which means that $\alpha F_S$ is reflected back to space and the remaining energy $(1 - \alpha)F_S$ is absorbed by the surface. The surface emits longwave radiation $I_S$, of which a fraction $\eta I_S$ is absorbed by greenhouse gases in the the atmosphere and the remainder $(1 - \eta)I_S$ escapes to space. The atmosphere emits longwave radiation $I_A$, of which a fraction $\beta I_A$ goes downward to the surface and the remaining fraction $(1 - \beta)I_A$ escapes to space. The three forcing terms $F_A, F_O, F_C$ represent atmospheric heat transport, ocean heat transport and vertical conduction/convection heat transport, respectively. Values of these and other parameters are given in Table 1.

## 2 An Energy Balance Model for Climate Change

The EBM is a simple two-layer model, with layers corresponding to the surface and atmosphere, respectively; see Fig. 1 which is based on (Payne et al. (2015); Trenberth et al. (2009); Wild et al. (2013)). The symbols in Fig. 1 are defined in the caption of Fig. 1 and in Table 1. In the EBM of Figure 1, the surface receives short-wave radiant energy $F_S = (1 - \xi_a - \xi_R)Q$ from the sun, where $Q$ is the incident solar radiation. and $\xi_A, \xi_R$ are the fractions of $Q$ absorbed by the atmosphere and reflected back to space, respectively. The values of $\xi_A$ and $\xi_R$ are obtained from (Trenberth et al. (2009); Wild et al. (2013)), see the Appendix

and Table 1. At the surface, a fraction $\alpha F_S$ is reflected back to space, where $\alpha$ is the surface albedo, and the remainer $(1 - \alpha)F_S$ is absorbed by the surface. The surface re-emits long wave radiant energy of intensity $I_S = \sigma T_S^4$ (Stephan-Boltzmann Law), upward into the atmosphere. The atmosphere contains greenhouse gases that absorb a fraction $\eta$ of the radiant energy $I_S$ from the surface, and the remainder $(1 - \eta)I_S$ escapes to space. The atmosphere re-emits radiant energy of total intensity $I_A$. Of this radiation $I_A$, a fraction $\beta$ is directed downward to the surface, and the remaining fraction $(1 - \beta)$ goes upward and escapes to

space.

Balancing the energy flows represented in Fig. 1 leads directly to the following dynamical system

$$c_S \frac{dT_S}{dt} = (1 - \alpha)(1 - \xi_A - \xi_R)Q + F_O + \beta I_A - \sigma T_S^4 - F_C \tag{1}$$

$$c_A \frac{dI_A}{dt} = F_A + \eta \sigma T_S^4 - I_A + F_C + \xi_A Q, \tag{2}$$

where (1) represents surface energy balance and (2) represents atmosphere energy balance. Here $c_S$ and $c_A$ are specific heat rate

constants derived in (Kypke (2019); Kypke and Langford (2020)) and listed in Table 1. There are three heat transport terms: $F_A$





is *atmospheric* heat transport and $F_O$ is *ocean* heat transport (both horizontally), and $F_C$ is *conductive/convective* heat transport, vertically from the surface to the atmosphere.

The first step of the analysis of system (1)(2) is a rescaling of temperature $T$ (in degrees Kelvin) to a new non-dimensional temperature $\tau$ with $\tau = 1$ corresponding to the freezing temperature of water ($T_R = 273.15$ K). Then all variables and parameters

in the system can be made non-dimensional by the scalings

$$\tau_A = \frac{T_A}{T_R}, \quad \tau_S = \frac{T_S}{T_R}, \quad q = \frac{Q}{\sigma T_R^4}, \quad f_O = \frac{F_O}{\sigma T_R^4}, \quad f_A = \frac{F_A}{\sigma T_R^4}, \quad f_C = \frac{F_C}{\sigma T_R^4}, \tag{3}$$

$$i_A = \frac{I_A}{\sigma T_R^4}, \quad \omega = \frac{\Omega}{T_R}, \quad s = \frac{\sigma T_R^3}{C_S} \cdot t, \quad \chi = \frac{C_S}{C_A \sigma T_R^3} = 54.26,$$

where $s$ is dimensionless time and $\chi$ is the dimensionless heat rate constant. The surface and atmosphere energy balance equations (1)(2) in non-dimensional variables are then

$$\frac{d\tau_S}{ds} = (1 - \alpha(\tau_S))(1 - \xi_A - \xi_R)q + f_O + \beta i_A - \tau_S^4 - f_C, \tag{4}$$

$$\frac{1}{\chi}\frac{di_A}{ds} = \left[f_A + \eta(\tau_S)\tau_S^4 - i_A + f_C + \xi_A q\right]. \tag{5}$$

The atmosphere in Fig. 1 is assumed to be a uniform slab, even though the actual atmosphere is not a uniform slab. However, the essential nonlinear processes in the atmosphere, which the model must capture, are the heating effects of the greenhouse gases $CO_2$ and $H_2O$. According to the Beer-Lambert law, the absorptivity of these gases is determined by their *optical depths*.

Therefore, in the model we substitute for optical depth in the slab, the values of optical depth that these gases would have in the International Standard Atmosphere (ICAO (1993)), which is a good approximation to the actual atmosphere. In this case, the rate of change of temperature with altitude is assumed to be a negative constant $-\Gamma$, called the *lapse rate*, see Table 1. The concentration of $CO_2$ is independent of temperature, but the concentration of $H_2O$ decreases with altitude as the temperature decreases, according to the Clausius-Clapeyron relation (Pierrehumbert (2010)). Then the optical depth of $H_2O$ is obtained by

integration of the Clausius-Clapeyron relation from the surface to the tropopause, as seen in (6). In this way, the simple two-layer model has greenhouse effects close to those of these two gases in the actual atmosphere. For further details see (Dortmans et al. (2019)). Thus, in the atmosphere equation (2) or (5), the total absorptivity $\eta$ due to greenhouse gases is determined as

$$\eta(\tau_S, \mu, \delta) = 1 - \exp\left[-\mu G_c - \delta G_{W2}\int_{\tau_S - \gamma Z}^{\tau_S}\frac{1}{\tau}\exp\left(G_{W1}\left[\frac{\tau - 1}{\tau}\right]\right)d\tau\right], \tag{6}$$

where $\mu$ is the concentration of $CO_2$ in the atmosphere, measured in molar parts per million, $\delta$ is the relative humidity of

water vapour ($0 \leq \delta \leq 1$), $\gamma = \Gamma/T_R$ is the nondimensionalized lapse rate (ICAO (1993)), and $Z$ is the tropopause height. (Since methane acts similarly to carbon dioxide as a greenhouse gas, it may be assumed that $\mu$ includes also the effects of methane.) Equation (6) is derived using fundamental laws of atmospheric physics: the Beer-Lambert law, the ideal gas law and the Clausius-Clapeyron equation, see (Dortmans et al. (2019)) for details. In equation (6),

$$G_c = \frac{1.52 k_c P_A}{10^6 g}, \quad G_{W1} = \frac{L_v}{R_W T_R} \quad \text{and} \quad G_{W2} = \frac{k_W \cdot T_R \cdot \rho^{sat}(T_R)}{\Gamma} \tag{7}$$





are physical constants determined in (Dortmans et al. (2019)) and $k_C$, $k_W$ are absorption coefficients for $CO_2$ and $H_2O$ respectively; see Table 1. Equation (6) determines the total greenhouse warming effects of $CO_2$ and $H_2O$, for given $\mu$ and $\delta$, and temperature $\tau_S$.

In the surface equation (1) or (4), $\alpha$ is the albedo of the surface ($0 \leq \alpha \leq 1$), and $\alpha$ depends strongly on temperature $\tau$ near the freezing point $\tau = 1$. Typical values of surface albedo are: 0.6–0.9 for snow, 0.4–0.7 for ice, 0.2 for crop land and

0.1 or less for open ocean. Therefore, in the high Arctic, as the ice-cover melts, the albedo will transition from a high value such as $\alpha_c = 0.7$ for snow/ice, to a low value such as $\alpha_w = 0.08$ for open ocean. Some authors have assumed this change in albedo to be a discontinuous step function (Dortmans et al. (2018)). However, all variables in this EBM have annually averaged values. As the Arctic thaws, the annual average albedo will transition gradually, over a number of years, from its high value for year-round ice-covered surface to its low value for year-round open ocean. Therefore, in this paper we use a smoothly varying

albedo function, which better models this gradual transition from high to low albedo, as the mean temperature rises through the freezing point ($\tau_S = 1$). It is modelled by the hyperbolic tangent function:

$$\alpha(\tau_S, \omega) = \frac{1}{2}\left( [\alpha_w + \alpha_c] + [\alpha_w - \alpha_c] \tanh\left( \frac{\tau_S - 1}{\omega} \right) \right). \tag{8}$$

Here the parameter $\omega$ controls the steepness of this switch function. Analysis of polar ice data in recent years confirms that this function gives a good fit to the decline of ice cover and albedo in the Arctic with $\omega = \Omega/T_R = 0.01$ (Dortmans et al. (2019)).

The dependence of Arctic Ocean sea ice thickness on surface albedo parametrization in models has been investigated in (Björk et al. (2013)), where alternative albedo schemes are compared. The nonlinear dependence of albedo on temperature, as in (8), has been shown to lead to *hysteresis* behaviour (Stranne et al. (2014); Dortmans et al. (2019)).

Previous papers (Dortmans et al. (2019); Kypke and Langford (2020)) used the EBM (4)(5) to investigate transitions in paleoclimates; in particular the Eocene-Ologocene Transition (EOT) and the Pliocene-Pleistocene Transition (PPT). In addition

to shedding light on the underlying causes of those transitions, the agreement achieved in those papers has served as a validation of the EBM, and of the fundamental hypothesis that bifurcations can occur in the Earth climate system.

## 2.1 Refinement of the Paleoclimate EBM to an Anthropocene EBM

Paleoclimate data are difficult to obtain and in general can only be inferred from proxy data. The situation is different for the Anthropocene. There is now an abundance of land-based and satellite climate data. Therefore, the EBM in this paper can

be refined to take advantage of the additional data. The Appendix details the changes made in the EBM, from that which was presented in (Dortmans et al. (2019); Kypke and Langford (2020)), to improve its accuracy for the Anthropocene. These changes do not change the fundamental behaviour of the EBM, including the existence of bifurcations, but they do make the numerical predictions more reliable. Table 1 of this paper may be compared with the corresponding Table 1 of (Kypke and Langford (2020)), to see how parameter values have been updated.





The total absorptivity $\eta$, given in (6) for paleoclimates, is modified, to reflect the fact that clouds absorb a fraction $\eta_{Cl}$ of the outgoing longwave radiation,

$$\eta(\tau_S) = 1 - (1 - \eta_{Cl}) \cdot \exp\left[ -\mu G_c - \delta G_{W2} \int_{\tau_S - \gamma Z}^{\tau_S} \frac{1}{\tau} \exp\left( G_{W1}\left[\frac{\tau - 1}{\tau}\right] \right) d\tau \right], \tag{9}$$

see the Appendix.

The vertical heat transport term $F_C$ has been modified to take into account both sensible and latent heat transport (Kypke (2019)) see the Appendix, where the following formula is obtained.

$$f_C = \frac{U}{\sigma T_R^4} \frac{P_0}{R_A(T_S - \Gamma Z)} \left( \frac{T_S}{T_S - \Gamma Z} \right)^{-g/R_A \Gamma}$$
$$\left[ c_p \, C_{DS} \, \Gamma Z + L_v \, C_{DL} \, \varepsilon \cdot \left( \frac{P^{sat}(T_R)}{P_0} \exp\left( \frac{L_v}{R_W}\left[ \frac{1}{T_R} - \frac{1}{T_S} \right] \right) \right) \cdot (1 - \delta) \right.$$
$$\left. + \frac{\delta \, L_v \, \Gamma Z}{R_W(T_S - \Gamma Z)^2} \left( \frac{P^{sat}(T_R)}{P} \exp\left( \frac{L_v}{R_W}\left[ \frac{1}{T_R} - \frac{1}{T_S - \Gamma Z} \right] \right) \right) \right]. \tag{10}$$

In equations (4)(5), at equilibrium (i.e. $\frac{d\cdot}{d\hat{s}} = 0$), the state variable $i_A$ is easily eliminated, leaving a single equation with a single state variable $\tau_S$,

$$0 = (1 - \alpha(\tau_S))(1 - \xi_A - \xi_{Cl})q + f_O + \beta f_A - f_C(1 - \beta) + \beta q \xi_A - \tau_S^4(1 - \beta \eta(\tau_S)). \tag{11}$$

## 2.2 Cusp Bifurcation in the EBM

In this subsection, we outline the proof that the cusp bifurcation, which was proven to exist in the Paleoclimate EBM (Kypke and Langford (2020)), in fact persists in this Anthropocene EBM (4)(5). Therefore, the conclusions of that paper carry over to this paper. Readers not interested in these mathematical details may skip this subsection.

The right hand sides of (4)(5), can be represented by a single vector function $F : \mathbb{R}^2 \times \mathbb{R}^4 \to \mathbb{R}^2$. Then an equilibrium point $(\bar{\tau}_S, \bar{i}_A)$ of (4)(5), at which $\frac{d\tau_S}{dt} = \frac{di_A}{dt} = 0$, is a solution of

$$F(\bar{\tau}_S, \bar{i}_A; \rho) = 0, \tag{12}$$

where $\rho$ represents four physical parameters that may be varied in the model,

$$\rho \equiv \{\mu, \delta, F_O, \omega\}. \tag{13}$$

See Table 1 for definitions of these parameters. Since the codimension of the cusp bifurcation is only two, there is some redundancy in the choice of these four parameters. For application to future climates, the parameter pair $(F_O, \mu)$ is of primary importance. Equilibrium points $(\bar{\tau}_S, \bar{i}_A)$ satisfying (12) have been computed in (Kypke (2019)). Having computed the equilibrium point $(\bar{\tau}_S, \bar{i}_A)$ satisfying (12), the system may be translated to the origin $(x, y) = (0, 0)$, in new state variables defined by

$$(x, y) \equiv (\tau_S - \bar{\tau}_S, i_A - \bar{i}_A),$$





and equations (4)(5) become

$$\frac{dx}{ds} = (1-\alpha)(1-\xi_A-\xi_{Cl})q + f_O - f_C + \beta(y+\bar{i}_A) - (x+\bar{\tau}_S)^4$$

$$\frac{dy}{ds} = \chi\left[f_A + f_C + q\xi_A + \eta(x+\bar{\tau}_S)^4 - (y+\bar{i}_A)\right]. \tag{14}$$

In order that 14 have a steady-state bifurcation at the equilibrium point $(0.0)$, the Jacobian $J$ of $F$ in (12) must have a zero eigenvalue $\lambda_1 = 0$ at that point. (A Hopf bifurcation is not possible in this system.) For stability, the second eigenvalue satisfies $\lambda_2 < 0$. The corresponding eigenvectors $\mathbf{e}_1, \mathbf{e}_2$ form an *eigenbasis*. A linear transformation takes $(x,y)$ coordinates to *eigenbasis coordinates* $(u,v)$, where $(x,y) = T(u,v)$, and the columns of $T$ are the normalized eigenvectors $\mathbf{e}_1, \mathbf{e}_2$ in (17), below. Then in
$(u,v)$ coordinates, the 2D system (14) becomes

$$\frac{du}{ds} = \frac{1}{\phi}\left[(1-\alpha)(1-\xi_A-\xi_{Cl})q + f_O + \beta f_A - (1-\beta)f_C + \beta\xi_A q + (1-\beta\eta)(u-kv+\bar{\tau}_S^4)\right] \tag{15}$$

$$\frac{dv}{ds} = \frac{1}{\phi}\left[-\ell[(1-\alpha)(1-\xi_A-\xi_{Cl})q + f_O] + (\ell+\chi)f_C + \chi f_A - (\ell\beta+\chi)[\ell u+v+\bar{i}_A] + (\ell+\chi\eta)[u-kv+\bar{\tau}_S^4]\right],$$

where

$$\ell = f'_{C0} + 4\eta_0\bar{\tau}_S^3 + \eta'_0\bar{\tau}_S^4, \quad k = \frac{\beta}{\chi}, \quad \phi = 1 + k\ell \tag{16}$$

and

$$\mathbf{e}_1 = \begin{pmatrix} 1 \\ \ell \end{pmatrix}, \quad \mathbf{e}_2 = \begin{pmatrix} -k \\ 1 \end{pmatrix}. \tag{17}$$

For more details, see (Kypke (2019); Kypke and Langford (2020); Kuznetsov (2004)).

If the *Centre Manifold Theorem* applies to (15), then there exists a flow-invariant centre manifold, tangent to the $u$-axis. The applicability of this theorem has been verified, and the centre manifold has been computed, for the Anthropocene EBM as was
done in (Kypke (2019); Kypke and Langford (2020)) for the paleoclimate EBM. Details are omitted here. A phase portrait for (15) in a neighbourhood of the cusp equilibrium point, together with a portion of this centre manifold (in red), is shown in Figure 2 in $(u,v)$ coordinates. In this figure, trajectories quickly collapse to the centre manifold around the equilibrium point $(0,0)$, as predicted by the Centre Manifold Theorem. The cusp equilibrium manifold for (15) in normal form is shown in Figure 3. Here $\beta_1, \beta_2$ are the normal form unfolding parameters for the cusp bifurcation. Note the co-existence of three
equilibrium points (two stable and the middle one unstable) inside the wedge-shaped region.

## 3 Anthropocene Climate Forecasts

In this section, the EBM is applied to future climates to investigate the possibility of climate bifurcations (or "tipping points") in the Anthropocene. The principal parameters chosen to be explored are carbon dioxide concentration $\mu$, ocean heat transport $F_O$ and relative humidity $\delta$.



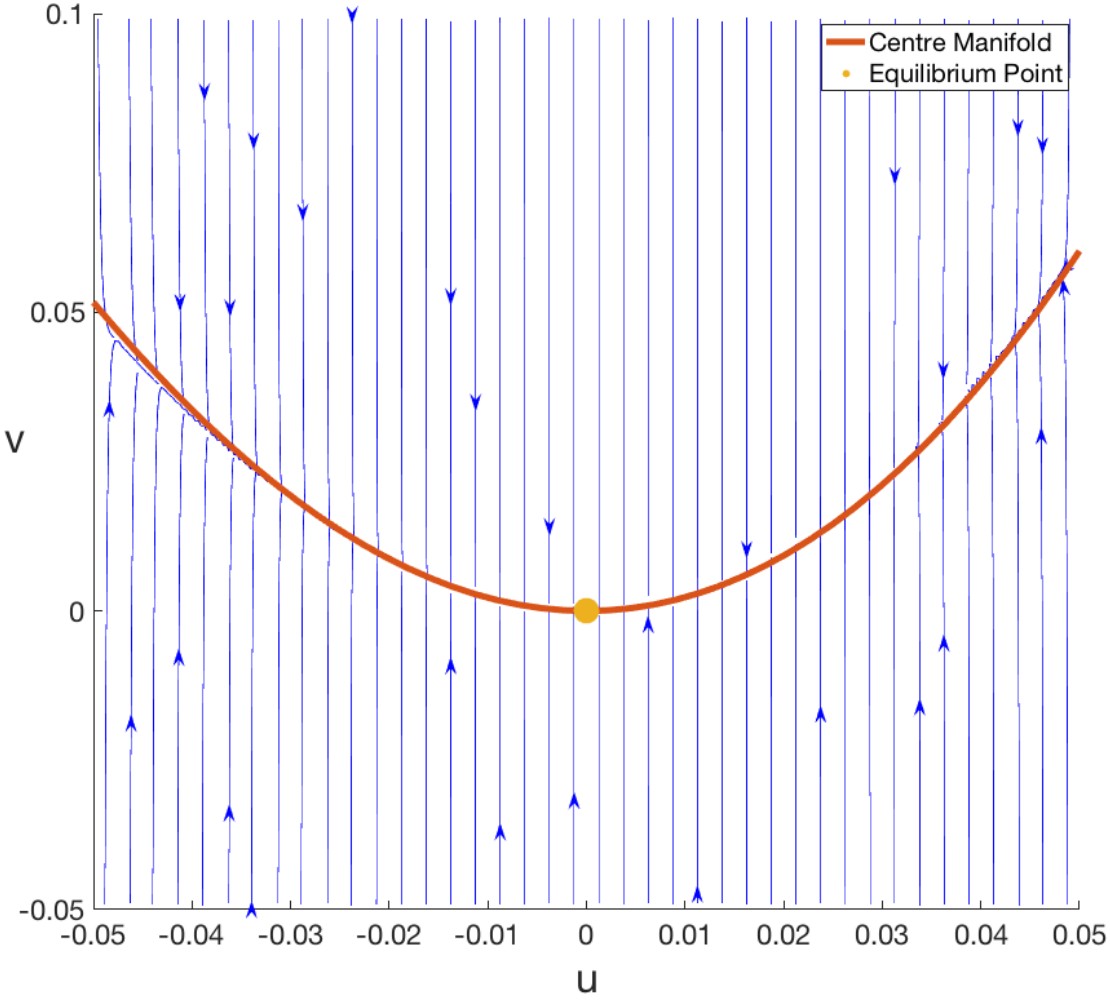

**Figure 2.** Phase portrait of system (15), together with a portion of the centre manifold (in red), in the (u,v) eigenbasis coordinates. Parameter values are those at the computed cusp point. The yellow dot marks the cusp equilibrium point. Note the rapid approach to the centre manifold from initial points not on the centre manifold.

Carbon dioxide production due to human activities has been well documented as a driver of climate change in the Anthropocene. Projections of future atmospheric $CO_2$ levels are available, under various future scenarios (IPCC (2013)). Ocean heat transport is a difficult quantity to predict, as many different factors influence the transport of heat to various regions of the world via the oceans. Changes in temperature can change ocean heat transport which in turn positively affects temperature. This is ocean heat transport feedback, which is explored in subsection 3.2.2. Similarly, the role of atmospheric heat transport



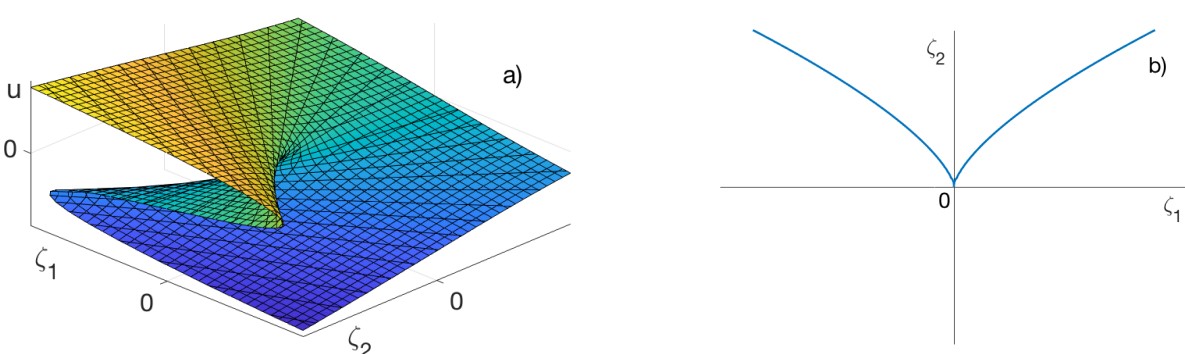

**Figure 3.** Cusp bifurcation diagram for the EBM in normal form coordinates. (a) Graph of the equilibrium surface with normal form unfolding parameters $(\zeta_1, \zeta_2)$. (b) Projection of this surface in 3D onto the $(\zeta_1, \zeta_2)$ plane. The blue semi-cubic parabola represents fold bifurcations, and it separates the $(\zeta_1, \zeta_2)$ plane into two open 2D regions. Inside the *wedge region*, that is between the two branches of the semi-cubic parabola, there exist three equilibrium solutions $u$, two stable and the middle one unstable. Outside of the semi-cubic parabola there exists only one unique equilibrium solution $u$ and it is stable.

feedback is investigated briefly in subsection 3.2.2. In addition, water vapour is a powerful greenhouse gas, with a positive feedback effect that is investigated in subsection 3.2.3.

The EBM is adapted to three separate regions, namely the Arctic, Antarctic and Tropics. A globally-averaged model also is considered, mainly for the purpose of determining the global Equilibrium Climate Sensitivity (ECS) of this EBM for comparison with that of other models, as reported in (IPCC (2013)); see Section 3.6.

## 3.1   Representative Concentration Pathways (RCP)

The IPCC has developed four Representative Concentration Pathways (RCP), which are used for projections of future carbon dioxide concentrations; see Box TS.6 in (IPCC (2013)). These RCPs are scenarios for differing levels of anthropogenic forcings on the climate of the Earth and represent differing global societal and political "storylines". The scenarios are named RCP 8.5, RCP 6.0, RCP 4.5 and RCP 2.6 after their respective peak radiative forcing increases in the 21st century. That is, in the

year 2100, RCP 8.5 will reach its maximum radiative forcing due to anthropogenic emissions of $+8.5$ W/m$^2$ relative to the year 1750. This scenario is understood as one where emissions continue to rise and are not mitigated in any way. RCP 6.0 corresponds to $+6.0$ W/m$^2$ and RCP 4.5 corresponds to $+4.5$ W/m$^2$ relative to 1750. These are stabilization scenarios, where greenhouse gas emissions level off to a target amount by the end of the century. Finally, RCP 2.6 corresponds to $+2.6$ W/m$^2$ in 2100, relative to 1750. This is a mitigation scenario, where strong steps are taken to eliminate the increase, and eventually

reduce, anthropogenic greenhouse gas emissions. Figure 4 shows the carbon dioxide concentrations, projected to the year



2500 in the IPCC scenarios for the four RCPs. The carbon dioxide increase is relatively moderate for RCPs 2.6 and 4.5, even decreasing eventually for RCP 2.6. The increase for RCP 6.0 is larger, and it is drastic for RCP 8.5.

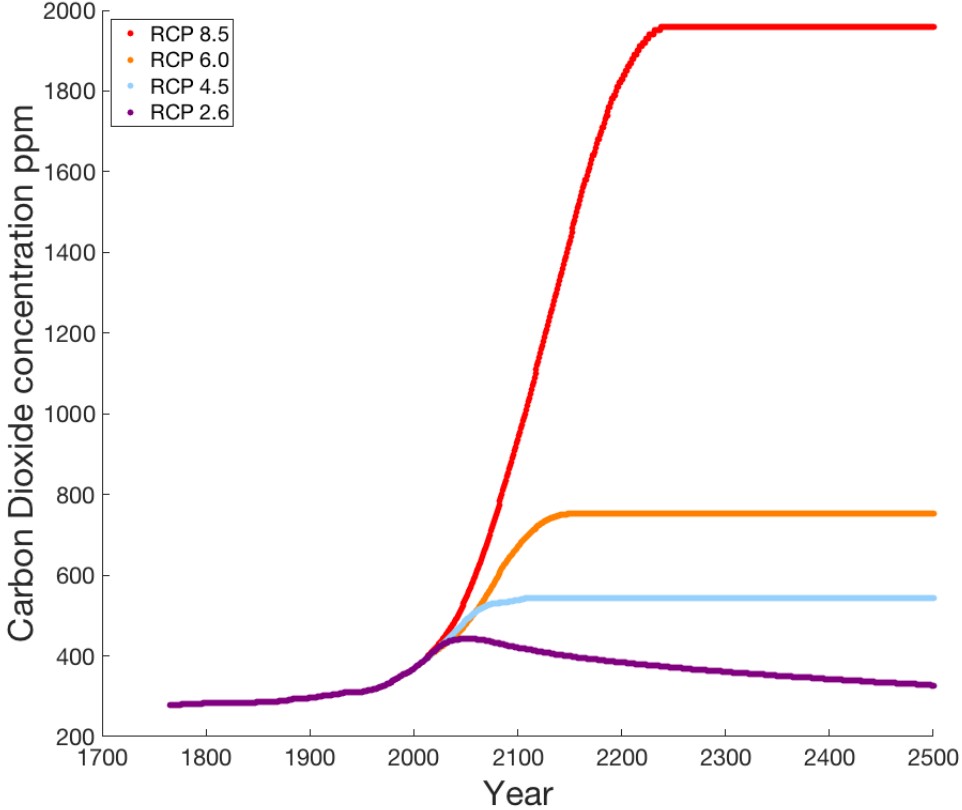

**Figure 4.** Carbon Dioxide concentration $\mu$, as projected by the IPCC for each of the four RCP hypothetical scenarios. This figure is generated from data at (IIASA (2019)).

The RCPs represent hypothetical forcing due to human activity up to the year 2100. Beyond 2100, they assume a "constant composition commitment", where emissions are kept constant, which serves to stabilize the scenarios beyond 2100 (IPCC (2013)). Of course, emissions could continue to increase, or be greatly reduced ("zero emissions commitment") at some point in the future. However, the constant emission commitment dataset provided in (IPCC (2013)), and shown in Figure 4, is what is utilized in this work. In the following sections we enter each of the four IPCC RCPs into the various versions of our EBM, and then let the climate evolve along each of the $CO_2$ pathways in Figure 4.





### 3.2 EBM for the Anthropocene Arctic

Figure 5(a) is the manifold of equilibrium points for the EBM parameterized by $(F_O, \mu)$, for the case of an Arctic climate under Anthropocene conditions. Figure 5(b) is the projection of this manifold onto the parameter plane, showing the fold bifurcation lines as boundaries between coloured regions. Parameter values are as in Table 1, with $F_A = 104$ W/m$^2$, and $\delta = 0.6$ taken as the nominal values for the modern Arctic throughout this Section, except in Figure 8. These figures were computed as in (Kypke and Langford (2020)). The cusp point, seen in Figure 3, still exists but is not visible in Figure 5, because it is outside

of the range of parameters included in the figure.

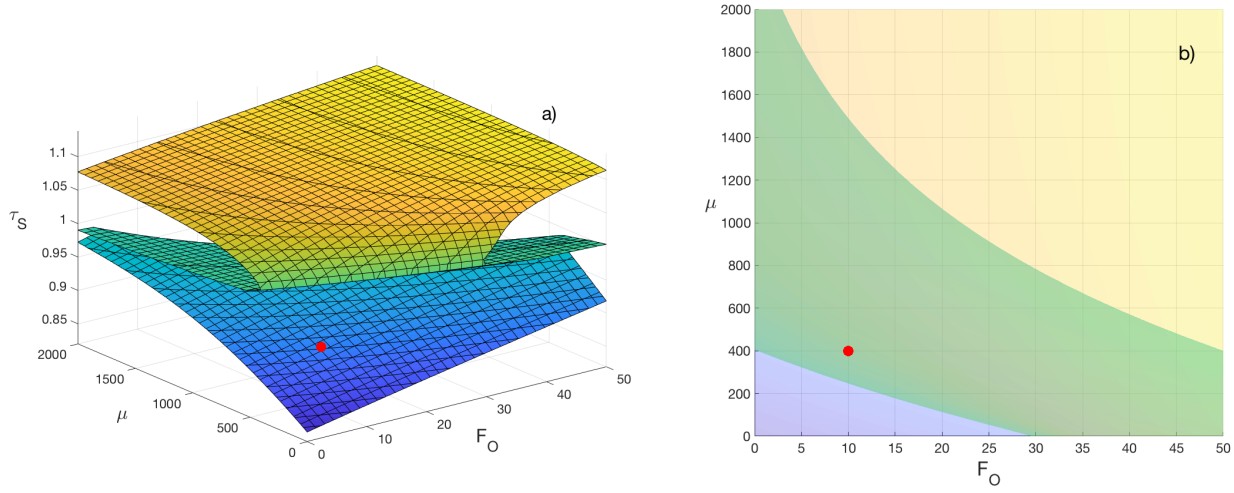

**Figure 5.** Energy Balance Model of the modern Arctic. Parameter values are as in Table 1. The red dots locate today's Arctic climate conditions. Ocean heat transport $F_O$ increases from 0 to 50 W/m$^2$ and carbon dioxide concentration $\mu$ increases from 0 to 2000 ppm. (a) 3D plot of equilibrium manifold. (b) Projection on the $(F_O, \mu)$ parameter plane.

In Figure 5(a), today's climate state lies on the lower (blue) portion of the equilibrium manifold, as shown by the red dot. The upper (yellow) portion represents a co-existing warm equable climate state, similar to the climate of Earth in the Pliocene and earlier. The intermediate (green) portion represents an unstable (and unobservable) climate state.

Similarly, in Figure 5(b), the blue area represents unique cold stable states, yellow represents unique warm stable states, and

the green region is the overlap region, between the two fold bifurcations, where both warm and cold states co-exist. Hence, on moving in the $(F_O, \mu)$ parameter space starting from the blue region, crossing the green region, and into the yellow region, there would be no observable change in climate on crossing from blue to green; however, crossing the boundary from green to yellow would cause a catastrophic jump from cold to warm stable climate states. Alternatively, if the $(F_O, \mu)$ parameter values moved from the yellow, through the green, into the blue region in Figure 5(b), there would be no abrupt change in climate state





on crossing the yellow/green boundary, but a sudden transition to a cold state would occur at the green/blue boundary. This behaviour is called *hysteresis*.

### 3.2.1 Arctic Climate for the 4 RCPs

The paramount question considered in this paper can now be phrased as follows. Can a bifurcation, leading to a warmer, more equable climate state, be expected in the EBM if it is allowed to evolve along one of the four RPCs in Figure 4? In

Figure 5(b), this bifurcation would correspond to crossing the line of fold bifurcations separating the green and yellow regions, on increasing $\mu$ and possibly $F_O$.

Figure 6 shows the increase in surface temperature in the Arctic region, using historical data from the year 1900 to the present, and then EBM projections up to the years 2100 and 2300, on each of the four RPCs, with constant $F_O = 10$ W/m$^2$, $F_A$ = 104 W/m$^2$ and $\delta = 0.6$. The temperature change shown is relative to the temperature of the EBM in the year 2000, which

was -28.42°C ($\tau_S = 0.8966$). This figure may be compared to the results seen in Figure AI.8 in (IPCC (2013)). While that Figure AI.8 provides surface temperature changes for the Arctic, for both sea and land separately, and only for the winter months of December - February, Figure 6 does not distinguish surface covering, and is an annual average value. Figure AI.9 in (IPCC (2013)) also provides the sea and land surface temperature changes during the summer months of June - August.

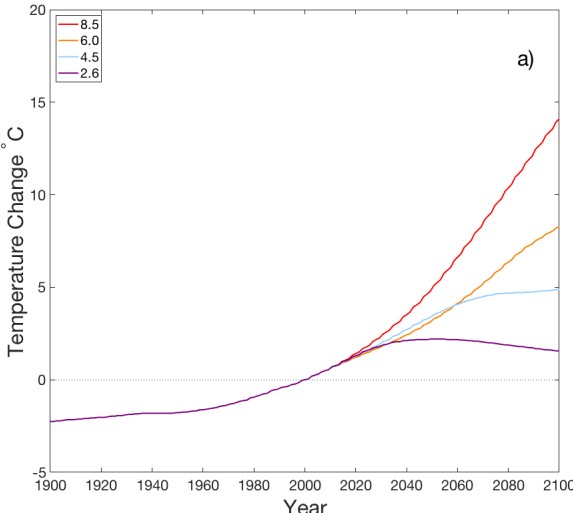
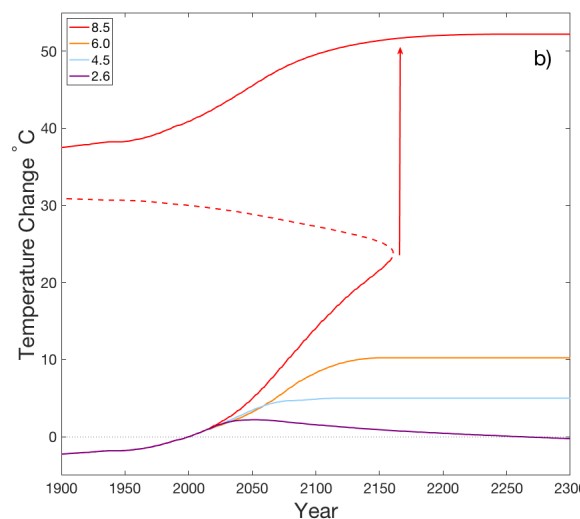

**Figure 6.** Arctic surface temperature change, relative to the year 2000 temperature of $-28.42$°C, for each of the four RCP's with constant $F_O = 10$ W/m$^2$ and with $\delta = 0.6$. In (a), the temperature change is projected to year 2100, and in (b) it is projected to year 2300, following the assumptions of the RCP pathways in Figure 4. Note the dramatic jump in temperature on RCP 8.5, resulting from a saddlenode bifurcation, predicted near the year 2160.





Figure 6(a) is in good agreement with the projections of (IPCC (2013)), if a weighted average of the sea and land temperature
changes are considered, and if the winter months are more representative of an annual value for the Arctic climate. However,
the projections in (IPCC (2013)) for Arctic region surface temperatures only extend to the year 2100. Figure 6(b) shows an
extension of the EBM forecasts to the year 2300, with exactly the same parameter values as in Figure 6(a). It exhibits a
dramatic bifurcation for RCP 8.5 near the year 2160. This bifurcation causes a drastic increase in temperature: the mean Arctic
temperature jumps by $+27.8°C$, to the new value of $+51.6°C$. Because of simplifications made in this EBM, these numbers
should not be taken literally as quantitatively accurate forecasts; however, the qualitative existence of a dramatic increase in
temperature due to a bifurcation must be taken seriously. The topological methods employed in this work ensure that the
bifurcation in this model is a mathematically persistent feature that will be manifest in all "nearby" models.

The other three RCPs show no such jump in Figure 6. However, it must be borne in mind that the IPCC assumptions (used
here) have all four RCPs levelling off to a target value by the end of this century, see Figure 4. This may be overly optimistic.

### 3.2.2   Ocean and atmosphere heat transport feedback

In Figure 6, the only forcing parameter that was changing was the $CO_2$ concentration (assumed due to anthropogenic forcing).
Now we incorporate changes in ocean and atmosphere heat transport, which may amplify the effects of increasing $CO_2$ alone.

There is evidence that ocean heat transport $F_O$ into the Arctic is increasing. For example, (Koenigk and Brodeau (2014))
project ocean heat transport above 70N to increase from 0.15 PW in 1860 to 0.2 and 0.3 PW in 2100, for RCP 2.6 and 8.5,
respectively. At the same time, they find in their model that atmospheric heat transport decreases slightly, from 1.65 PW in
1850 to 1.6 PW (1.5 PW) for RCP 2.6 (RCP 8.5). These authors found that, in a stable climate state that ensures global energy
conservation, $F_O$ and $F_A$ tend to be out of phase; see for example the coupled climate model in (Koenigk and Brodeau (2014)).
However, (Yang et al. (2016)) show that in a more realistic situation when the climate is perturbed by both heat and freshwater
fluxes, the changes in $F_O$ and $F_A$ may be in-phase. We assume the latter situation in this paper, see Figure 7(b).

In our model, climate forcings $F_O$ and $F_A$ are expressed as power per unit area $W/m^2$, determined as follows (see Table 2).
First, the surface area of the Arctic region is estimated. The Arctic is taken to be the surface of the Earth above the 70th parallel;
as such the surface area is

$$\text{Arctic Surface Area} = 2\pi R^2(1 - \cos\theta), \tag{18}$$

where $R$ is the radius of the Earth, 6371 km, and $\theta$ is $90°$ minus the latitude. Hence, the surface area of the Earth above $70°$
is approximately $1.538 \times 10^{13}$ $m^2$. This leads to atmospheric and ocean heat fluxes into the Arctic as summarized in Table 2.
Because the change in $F_A$ is small relative to the changes in $\mu$ and $F_O$, $F_A$ is kept constant at an intermediate value of 104 $W/m^2$
in Figures 5 to 7(a).

Figure 7(a) shows the change in Arctic surface temperature (relative to the year 2000 temperature of $-28.42°C$) for the four
RCPs with the ocean heat flux $F_O$ increasing linearly on each RCP until the year 2100, as projected in (Koenigk and Brodeau
(2014)), using their data for $F_O$ in Table 2, but with constant $F_A = 104$. Beyond the year 2100, until 2300, the ocean heat
transport $F_O$ is held constant at its 2100 value. In this scenario, the onset of the jump (via a fold bifurcation) to a warm equable





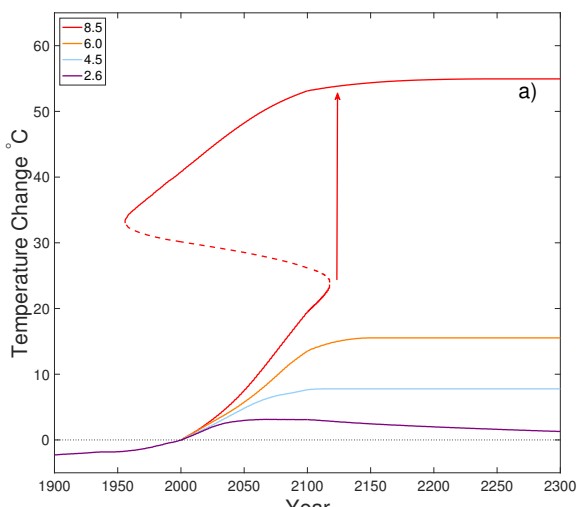

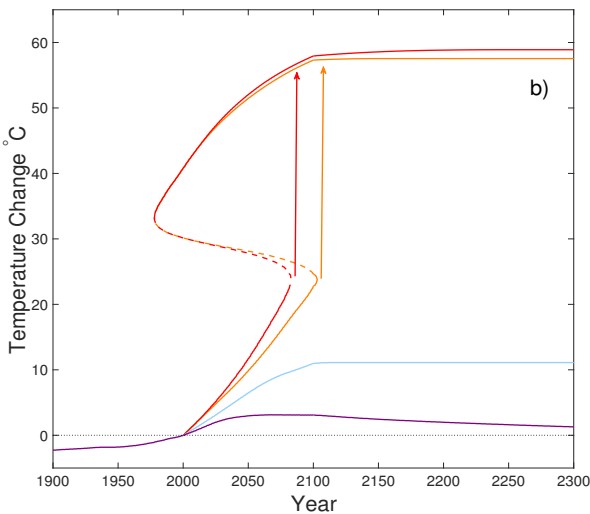

**Figure 7.** Arctic surface temperature change projected to year 2300 (relative to year 2000 temperature of $-28.42°$C), with linearly increasing ocean heat transport $F_O$, interpolating the data in (Koenigk and Brodeau (2014)) see Table 2. (a) With constant $F_A = 104$, the jump in temperature for RCP 8.5 now occurs nearly 40 years earlier than for the case of constant $F_O$ shown in Figure 6. (b) The same as (a) except that in addition to increasing $F_O$, the atmosphere heat transport $F_A$ also increases, linearly from 104 to 129 W/m$^2$. Now bifurcations occur on *both* RCP 8.5 and 6.0.

climate is advanced dramatically. The bifurcation for RCP 8.5 occurs in the year 2118, more than 40 years earlier than was the case with a constant $F_O$ (Figure 6). The temperatures associated with the jump between two stable states are from $+23.8°$C in 2118 to $+53.1°$C in the year 2119, a sudden increase of 29.9°C.

Figure 7(b) shows the same scenario as in (a), with increasing $\mu$ and $F_O$, but with the atmospheric heat transport $F_A$ also increasing, linearly from 104 W/m$^2$ in the year 2000 to 129 W/m$^2$ in 2100, and constant thereafter. In this case, the bifurcation occurs earlier for RCP 8.5, and a new bifurcation occurs for RCP 6.0. Both of these changes make mitigation more challenging.

### 3.2.3   Water vapour feedback

Overall, water vapour is known to be the most powerful greenhouse gas in the atmosphere (Dai (2006); Pierrehumbert (2010);
IPCC (2013)). Warming of the surface causes evaporation of more water vapour, which causes further greenhouse warming and a further rise in surface temperature. Thus, water vapour amplifies the warming due to other causes. This is called water vapour feedback. Empirical studies such as (Dai (2006)) show that the increase in surface specific humidity $H$ with surface temperature $T$ is close to that predicted by the Clausius-Clapeyron equation as in (6) (outside of desert regions). The relative humidity, RH or $\delta$, changes little with surface temperature, even as the specific humidity $H$ increases (Serreze et al. (2012)). For
paleoclimates, (Jahren and Sternberg (2003)) have described an Eocene Arctic rain forest with RH estimated to be $\delta = 0.67$.





Modern data, from a variety of sources, suggest similar values of Arctic RH. For example, (Shupe et al. (2011)) report Arctic RH at the surface over 0.7 and atmospheric RH at 2.5 km altitude near 0.6, with relatively small seasonal and spatial variations.

Therefore, in the EBM (1)(2), it is assumed that the greenhouse warming effect of water vapour is determined by the Clausius-Clapeyron relation as in equation (6) and the RH $\delta$ remains constant. We assumed a value of $\delta = 0.60$ for the Arctic atmosphere in the previous section.

The Clausius-Clapeyron equation tells us that, below the freezing temperature ($\tau_S = 1$) the concentration of water vapour is very low and therefore it has very little greenhouse effect. However above freezing, if a source of water is available (e.g. oceans), then the concentration of water vapour and its greenhouse warming effect increase rapidly. This is shown clearly in Figure 8, where the three curves show different levels of relative humidity $\delta$, but all assume that $CO_2$ follows RPC 8.5. Here, the reference temperatures in year 2000 are as follows: Red curve -28.13°C, Green curve -28.95°C, Blue curve-29.50°C. On each of the curves of Figure 8, portions with negative slope are unstable, while portions with positive slope are asymptotically stable (in the sense of Liapunov). Bistability (the coexistence of two stable solutions) occurs sooner when water vapour is present. The lower blue curve with $\delta = 0$ shows no thawing ($\tau_S < 1$) in this range.

### 3.2.4 Anthropocene Arctic EBM summary

This EBM for the Anthropocene Arctic predicts a bifurcation, producing catastrophic warming of the Arctic, if $CO_2$ emissions continue to increase unmitigated, even while ocean and atmosphere heat transport remain unchanged. The amplifying effects of ocean and atmosphere heat transport can make this abrupt climate change become even more dramatic, and occur even earlier. Water vapour feedback further intensifies global warming. However, the EBM predicts that $CO_2$ mitigation strategies, if introduced soon enough, may avert the drastic consequences of this bifurcation.

Further work on Anthropocene Arctic climate modelling will include the effects of other positive feedback mechanisms, for example the greenhouse effects of the methane and $CO_2$ that will be released as the permafrost thaws in the Arctic, and the Hadley cell feedback that will occur as global circulation patterns change. With such additional amplification in the Arctic taken into account, and no mitigation strategies in place, a saddlenode bifurcation to a warmer Arctic climate state may ocurr even earlier than predicted by the present model.

### 3.3 EBM for the Anthropocene Antarctic

The Antarctic climate differs from the Arctic climate in one major way: It is affected by the Antarctic Circumpolar Current (ACC), which flows freely west to east around Antarctica in the southern ocean, unimpeded by continental barriers. The ACC blocks the poleward heat transport from the warm oceans of the southern hemisphere (Hartmann (2016)). Hence, $F_O$ is restricted between 0 and 20 W/m$^2$ in the Antarctic EBM. Additionally, the cold surface albedo $\alpha_C = 0.8$ is greater for the Antarctic than the Arctic, because the snow and ice that covers the continent is more pure than that in the Arctic. Cloud albedo is reduced, with a value of 7% as opposed to the value of 12.12% for the Arctic (Pirazzini (2004)). Atmospheric heat transport is $F_A = 94$ W/m$^2$, as determined in (Zhang and Rossow (1997)). Finally, the Antarctic region is much drier than the Arctic, hence a relative humidity of $\delta = 0.4$ is used.


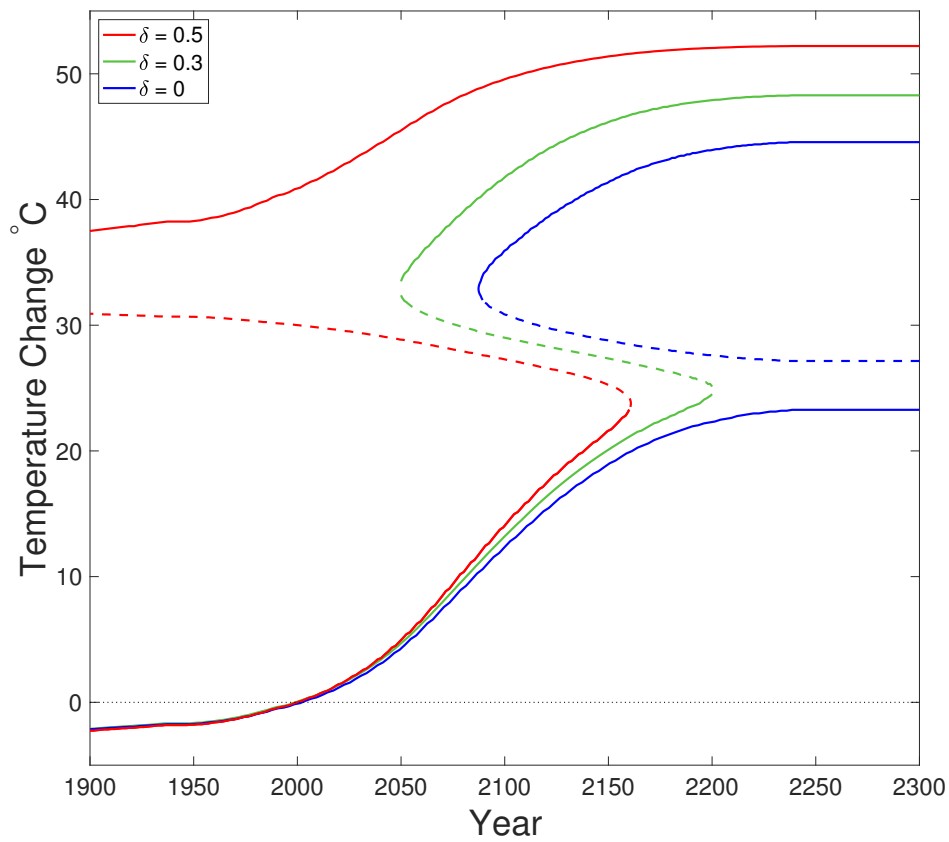

**Figure 8.** Arctic surface temperature change (relative to the year 2000 temperatures at approximately 29°C, see text) projected to year 2300, with increasing relative humidity of water vapour, $\delta = 0.0$, $0.3$, $0.6$, and with fixed $F_O$, $F_A$. On all three curves, $CO_2$ is increasing in time according to RCP 8.5. The red curve is essentially the same as that shown in Figure 6 with $\delta = 0.6$. The blue and green curves are RCP 8.5, but with water vapour fixed at $\delta = 0.0$ and $\delta = 0.3$, respectively. For temperatures significantly below freezing, water vapour makes little contribution to temperature change. However, above freezing ($\tau_S > 1$), the greenhouse warming effect of water vapour is dramatic.

For the Antarctic, Figure 9(a) is the equilibrium manifold for the energy balance model parameterized by $(F_O, \mu)$, and
Figure 9(b) is the projection of the fold bifurcations onto the parameter plane. In Figure 9(b), the yellow area represents a warm stable climate, the blue area a cold stable climate, and the green area the overlap (between the two fold bifurcations). It can be seen in Figure 9 that a bifurcation from a cold state to a warm state cannot occur for an ocean heat transport value of less than $F_O = 12$ W/m$^2$ and a carbon dioxide concentration less than $\mu = 3000$ ppm.

Figure 10 shows the temperature change, following each of the RCP curves for the Antarctic, extended to the year 2300. The
reference temperature is $-33.38$°C, for the year 2000, and the value of ocean heat transport into the Antarctic is assumed to be

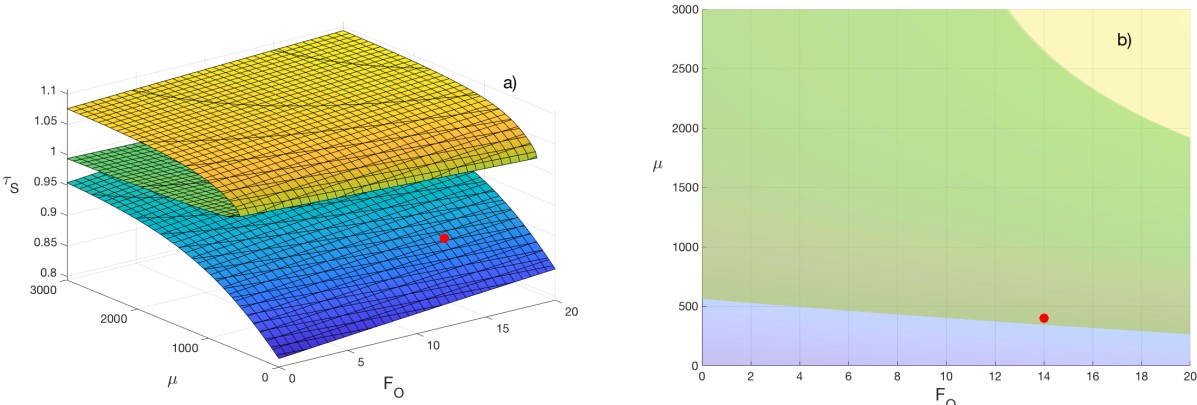

**Figure 9.** Energy Balance Model for the Antarctic. (a) 3D equilibrium manifold, (b) Projection of fold bifurcations The red dot locates today's climate conditions.

$F_O = 14$ W/m$^2$, an annual mean for the sea-ice zone (approximately 70°S) of the Antarctic, as determined in (Wu et al. (2001)). This scenario does not exhibit a bifurcation from a cold climate state to a warm state. This suggests that for the Antarctic, a change in $\mu$ alone is not sufficient to cause a hysteresis loop to exist, between two coexisting stable states, in the context of modern and near-future carbon dioxide concentrations.

Figure 11 presents the Antarctic model for values of $F_O$ that increase with time as $\mu$ does. The value of $F_O$ is kept constant at 5 W/m$^2$ until the year 2000, after which point it increases linearly up to the year 2100, where it has a value of $F_O = 20$ W/m$^2$, after which it is held constant again. This increase might represent an increase in sea levels, caused by thawing of the Arctic, and subsequent increase in ocean heat transport (Koenigk and Brodeau (2014)). The first thing to notice is that a hysteresis loop now exists. With increasing CO$_2$ on RCP 8.5, there is a a jump between stable states, from +29.1°C in 2224 to +56.5 °C

in the year 2225. The return bifurcation, from warm to cold, also may be seen in Figure 11. The "cold-to-warm" bifurcation occurs at a later time than in the Arctic, and at a greater temperature. The difference in the Antarctic bifurcations, as compared to the Arctic, is due to the difference in ocean heat transport and ice albedos. The difference could be larger if other factors are taken into account, for example that the Antarctic has thicker ice, hence more heat is required to melt enough ice to cause a change in albedo. This could be represented with a different value in $\omega$: a smaller value for the tanh switch function smoothness

represents a greater temperature required for the function to switch from a cold albedo value to a warm one.

### 3.4  EBM for the Anthropocene Tropics

Next, the EBM is adapted to model the climate of the Tropics by choosing parameter values that are annual mean, zonally averaged values at the equator. This gives insolation $Q = 418.8$ W/m$^2$ and relative humidity $\delta = 0.8$. Heat transports $F_A = -38$ W/m$^2$ and $F_O = -39$ W/m$^2$ are both negative, because heat is transported away from the equator towards the poles (Hartmann

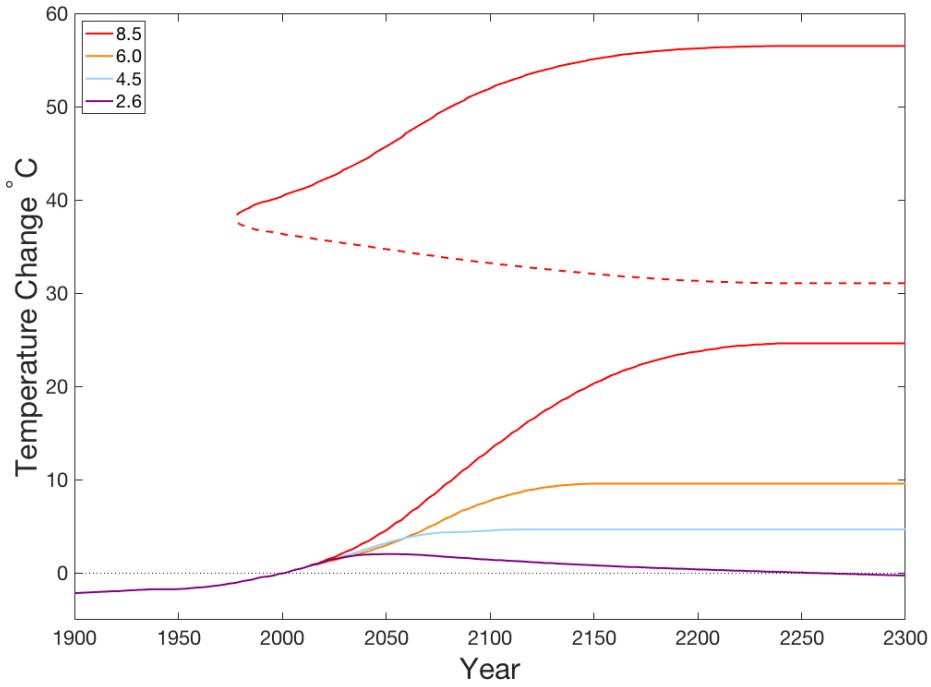

**Figure 10.** Antarctic surface temperature change projected to year 2300 (relative to year 2000 temperature of $-33.38°$C), on each of the four RCPs, and with constant $F_O = 14$ W/m$^2$

(2016)). The shortwave cloud cooling, i.e. the albedo of the clouds, is also greater in the Tropics, and the surface has a lower albedo. The value used is determined in the Appendix from the global energy budgets of (Trenberth et al. (2009); Wild et al. (2013)), a value of 22.35%.

As the Tropics have annual average temperatures well above the freezing point of water, ice-albedo feedback is absent in the Tropics and a bifurcation from a cold stable state to a warm state can not occur under Anthropocene conditions. However,

if forced to low $F_O$, $F_A$ values and very low carbon dioxide levels, the climate state known as "snowball Earth" (Pierrehumbert (2010)) is a possibility. That scenario is not explored in this paper.

The large relative humidity of $\delta = 0.8$ in the tropics serves to mitigate the radiative forcing of increasing $CO_2$. Water vapour is a much more effective greenhouse gas than carbon dioxide, so for a climate that contains more water vapour (the product of greater relative humidity and warmer temperatures), the effect of an increase in carbon dioxide is smaller compared to the case

of a climate where water vapour is less abundant (Pierrehumbert (2010)). The total atmospheric longwave absorption $\eta$ is as given in equation (9); so in a region where the water vapour content is high, $\eta_W$ (and thus total absorptivity $\eta$) will be almost "maxed out" at 1. Hence the total absorptivity is dominated by the contribution due to water vapour, and an increase in $CO_2$ concentration will have little additional greenhouse warming effect.


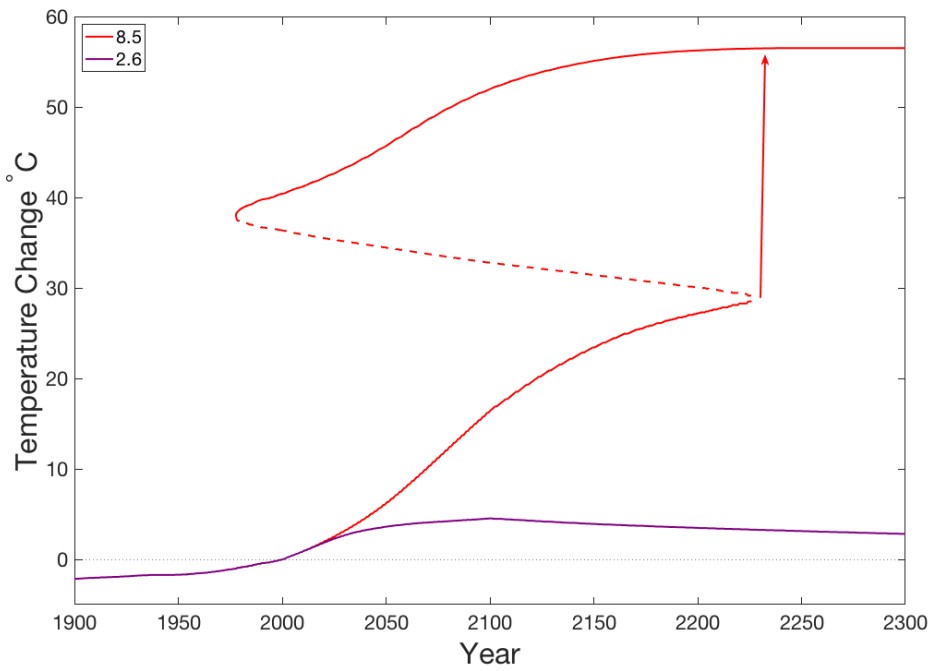

**Figure 11.** Antarctic surface temperature change projection to year 2300 relative to year 2000 with increasing $F_O$, see text.

Figure 12 reveals relatively low increases in temperature for the Tropics compared to the Poles. Both the absence of a
bifurcation, and the mitigation due to large existing water vapour greenhouse forcing, taken together cause the temperature
increase relative to the year 2000 to be less than 2°C, in all four RCP scenarios.

### 3.5 EBM for Globally Averaged Temperature

Changes in globally averaged temperature can be modelled more easily than changes in regional temperatures, due to the fact
that, in a globally averaged equilibrium model, overall net horizontal transport of energy, by the oceans and the atmosphere,
are both zero. Thus, the two-layer EBM (4) (5), globally averaged with $F_O = 0$ and $F_A = 0$, becomes as follows.

$$\frac{d\tau_S}{d\hat{s}} = (1 - \alpha)(1 - \xi_A - \xi_{Cl})q - f_C + \beta i_A - \tau_S^4 \tag{19a}$$

$$\frac{di_A}{d\hat{s}} = \chi[f_C + q\xi_A + \eta \tau_s^4 - i_A]. \tag{19b}$$

Here $\alpha$ is as defined in equation (8), $\eta$ is as in (9) and $f_C$ is as in (10). Parameters $\xi_A, \xi_{Cl}$ are as in Table 1 and Section 2.1.

Figure 13 shows the change in globally averaged equilibrium surface temperature, relative to the year 2000 global average
surface temperature ($\tau_S = 1.063$, 17.21°C), determined by the EBM (19) to the year 2300. It is assumed that $CO_2$ evolves
with time along each of the four RCPs defined in (IPCC (2013)) and displayed in Figure 4. The other parameters, assumed

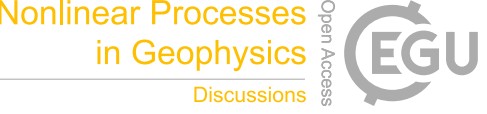

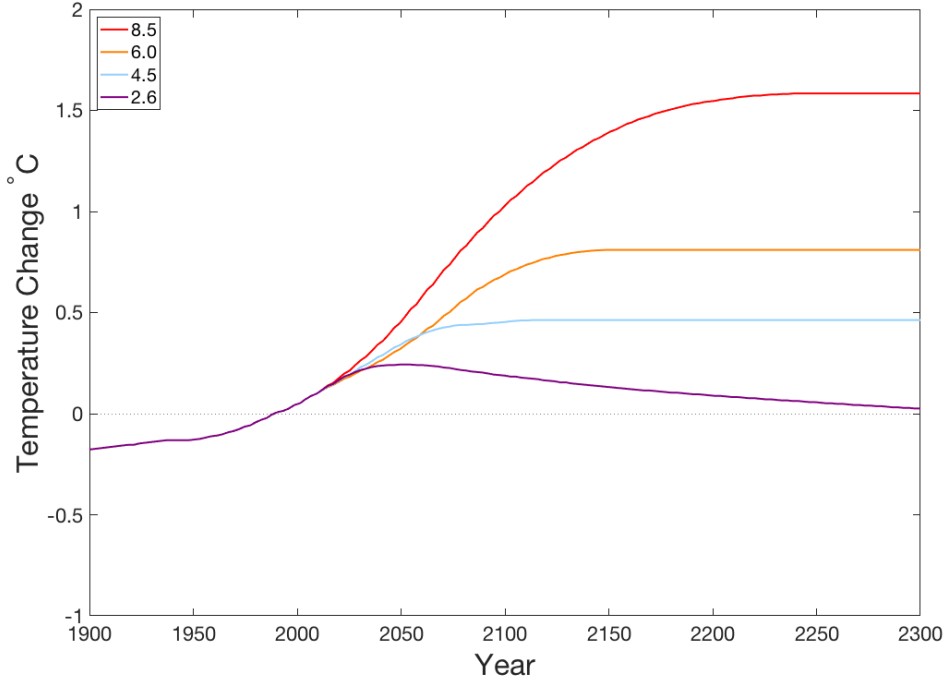

**Figure 12.** Tropical surface temperature changes for the four RCP scenarios, forecast to year 2300 relative to year 2000 (27.59°C), with constant $F_O$ and $F_A$. The temperature change is relatively small in the Tropics.

constant, are as follows. The global relative humidity $\delta$ is fixed at a value of 0.74. This is determined from (Dai (2006)), where it lies at the lower end of a range of averages. Surface albedo is highly variable regionally, so a global average was calculated from (Wild et al. (2013)), much like the atmospheric shortwave absorption and the cloud albedo. From Figure 1 of (Wild et al. (2013)), of the global average solar radiation of 185 W/m² that reaches the surface, a portion 24 W/m² is reflected. Thus, the global average surface albedo is $\frac{24}{185} = 0.13 = 13\%$. The values for cloud albedo and atmospheric shortwave radiation are calculated as follows. The global average incident solar radiation $Q$ at the top of the atmosphere is 340 W/m², of which $\frac{100-24}{340} = 0.2235 = 22.35\%$ is reflected by clouds, and $\frac{79}{340} = 0.2324 = 23.24\%$ is absorbed by the atmosphere. The Planetary Boundary Layer (PBL) altitude is 700 m (Ganeshan and Wu (2016)). Finally, the wind speed $U$, for the purposes of sensible and latent heat transport, is 6 m/s (Nugent et al. (2014)).

The changes in surface temperature from the year 2000 reference value, as shown in Figure 13, agree well with the changes predicted in the IPCC report; namely, 7.8°C for RCP 8.5, 2.5°C for RCP 4.5, and 0.6°C for RCP 2.6 (IPCC (2013)). (A value for RCP 6.0 was not given in (IPCC (2013)).)





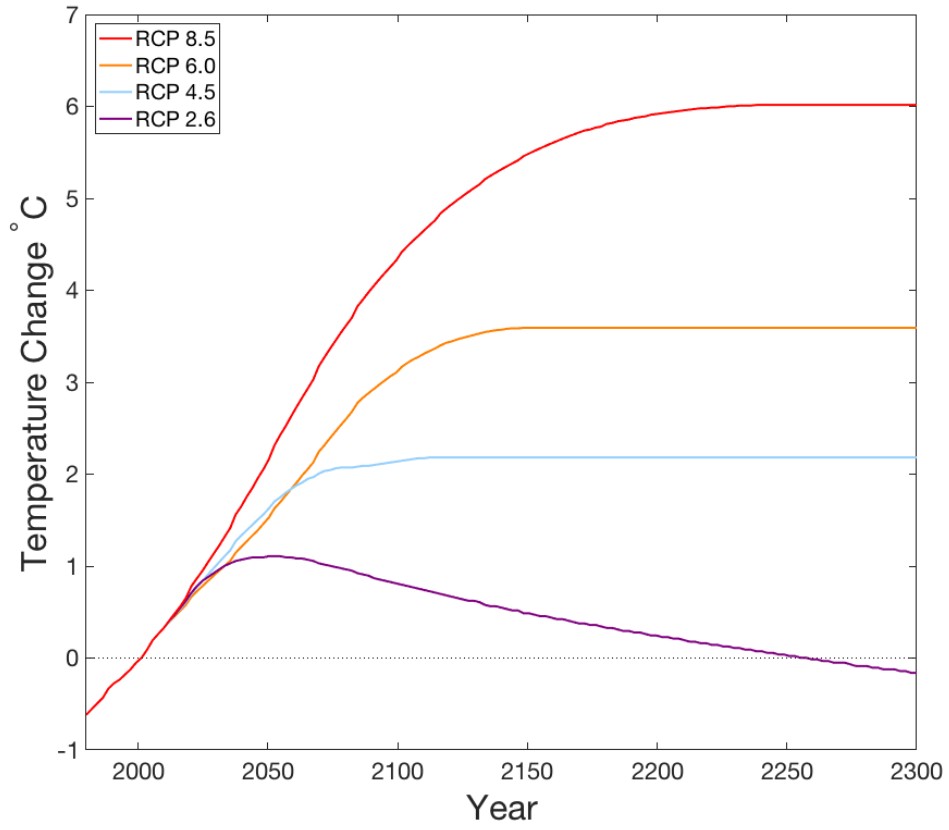

**Figure 13.** Change in globally averaged surface temperature, relative to year 2000 global average temperature of 17.21 °C, predicted to the year 2300 for each of the four RCPs.

## 3.6 Equilibrium Climate Sensitivity

*Equilibrium climate sensitivity* (ECS) is a useful and widely-adopted tool used to estimate the effects of anthropogenic forcing in a given climate model. The ECS of a model is defined as the change in the globally averaged surface temperature, after equilibrium is obtained, in response to a doubling of atmospheric $CO_2$ levels (IPCC (2013); Knutti et al. (2017); Priostosescu and Huybers (2017)). The starting carbon dioxide concentration is that of the preindustrial climate, taken to be $\mu = 270$ ppm. The doubled value is then $\mu = 540$ ppm. Since the Earth has not yet experienced a doubling of $CO_2$ concentration since the

industrial revolution, these numbers cannot be verified. Calculation of the global ECS for the EBM of this paper facilitates comparisons with other climate models as reported in (IPCC (2013)).





### 3.6.1 ECS for the globally averaged EBM

Table 3 gives both the non-dimensional $\tau_S$ and the degree Celsius temperature values for the $\mu = 270$ climate, the $\mu = 540$ climate, and the temperature difference. This difference is the ECS of the global EBM of this paper.

For the models used in (IPCC (2013)), ECS values lie within a *likely* range of 1.5 °C to 4.5 °C. Values of less than 1 °C are deemed to be *extremely unlikely*, and greater than 6 °C are *very unlikely* (IPCC (2013)). The value of 4.55 °C calculated for this global EBM lies just above the *likely* range, but is still well below the *very unlikely* boundary. Recent work gives evidence that statistical climate models based on historical data tend to lie on the lower end of *likely* ECS values, with a range of 1.5 °C to 3 °C, whereas nonlinear GCMs tend to have larger ECS values (Priostosescu and Huybers (2017)). Therefore, as the global

EBM presented in this paper is nonlinear and is based on geophysics rather than statistical data, it may be expected to fall on the side of larger ECS values.

### 3.6.2 Regional ECS values

Local ECS values may be determined for each of the three regional models, for the Arctic, Antarctic and Tropics, as defined in Sections 3.2 to 3.4. These values are given in Table 4. In all cases, $F_O$ values are kept constant at their minimal values: 10

W/m$^2$ for the Arctic, 14 W/m$^2$ for the Antarctic, and -39 W/m$^2$ for the tropics. The regional ECS values are high, 8.0 and 7.5 °C respectively for the Arctic and Antarctic, and low, 1.1 °C for the Tropics. Although the Earth has not yet experienced a doubling of $CO_2$ concentrations since the industrial revolution, these ECS values are consistent with observations to date.

## 4   Conclusions

The analysis of this paper presents a mathematical proof that a bifurcation can occur in an energy balance model (EBM) for

the anthropocene climate, which has been constructed from the fundamental nonlinear processes of atmospheric physics. This bifurcation is most likely to occur in the Arctic climate. It would lead to catastrophic warming, if increases in atmospheric $CO_2$ continue on their current pathway. However, if the increase in atmospheric $CO_2$ is mitigated sufficiently, this bifurcation, causing catastrophic climate change, can still be avoided. Climate changes in the Arctic, Antarctic and Tropics are compared. The globally averaged equilibrium climate sensitivity (ECS) of the EBM is shown to be at the high end of the range considered

in (IPCC (2013)).

    Future work will strengthen the conclusions of this paper by extending this simple EBM to more comprehensive Earth system models, which still allow rigorous bifurcation analysis to be performed. In the first generalization, the two-layer model will be replaced with a column model, with the atmosphere extending continuously from the surface to the tropopause. The ICAO International Standard Atmosphere will be replaced with a Schwarzschild radiation model of the atmosphere, which will

determine the lapse rate from the solution of a two-point boundary value problem. This Schwarzschild column model will be used to study, in addition to the positive feedback processes of this paper, the amplifying effects of permafrost feedback in the Arctic.



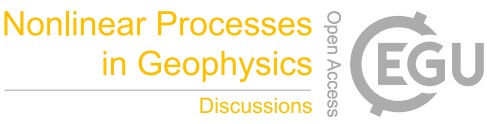

The next generalization will be to a 3D model, with Navier-Stokes-Boussinesq PDEs representing the convective atmosphere in a spherical shell, as presented in (Lewis and Langford (2008); Langford and Lewis (2009)), but with the addition of an energy

balance constraint like that in the Schwarzschild column model, which will determine implicitly the surface temperature. The relationship between polar amplification and Hadley cell expansion will be explored with this model.

Later, guided by these results, the climate bifurcations found in these analytical models will be sought in an open source General Circulation Model. This hierarchy of models is expected to add credibility to the prediction that the Earth's climate system may exhibit dramatic topological changes (bifurcations) in the Anthropocene.

**Appendix. Determination of Parameter Values in the EBM**

The determination of the physical parameters appearing in the EBM (4)(5) is summarized here. In most cases, these parameters were determined in earlier papers of the authors (Dortmans et al. (2019); Kypke and Langford (2020)). The starting point is the scaled, two-dimensional equations (4)(5) in Section 2.1, with $\alpha$ given by (8) and $\eta$ given by (9).

First consider the incoming solar radiation Q. A fraction $\xi_A$ is absorbed by the atmosphere and a fraction $\xi_R$ is reflected by

the clouds, as seen in Figure 1 and equations (4)(5). These fractions were determined in (Kypke (2019)) and Appendix A1 of (Dortmans et al. (2019)), using data for the global average energy budget described in (Trenberth et al. (2009); Wild et al. (2013); Kim and Ramanathan (2012)). The globally averaged values were $\xi_A = 0.2324$ and $\xi_R = 0.2235$ as listed in Table 1. For the polar regions, the albedo of clouds is less than elsewhere. Using data collected in the Surface Heat Budget of the Arctic (SHEBA) program (Intrieri et al, (2002); Shupe and Intrieri (2003)), the polar value of $\xi_R = 0.1212$ was determined in (Kypke

440 (2019)).

Clouds in the atmosphere have a second main effect, the absorption of a fraction $\eta_{Cl}$ of the longwave radiation outgoing from the surface of the Earth (Hartmann (2016)). This effect warms the Earth's atmosphere. Through data in the SHEBA program, the longwave cloud forcing in the Arctic was found to be 51 W/m$^2$ in the paper (Shupe and Intrieri (2003)). Using this SHEBA data, Kypke (2019) determined the fraction $\eta_{Cl} = 0.255$, as used in (9), see Table 1.

The absorption coefficients $k_C$ for carbon dioxide and $k_W$ for water vapour in the atmosphere (see Table 1), were calculated using an empirical approach, based on the modern-day global energy budget (Trenberth et al. (2009); Wild et al. (2013)). Figure 1 in (Trenberth et al. (2009)) provides the global mean surface radiation as 396 W/m$^2$, along with an atmospheric window of 40 W/m$^2$. This atmospheric window, $\frac{40}{396} \approx 0.1$, is then equal to $1 - \eta$. (Schmidt et al. (2010)) provide percentage contributions of carbon dioxide and water vapour and clouds in an all-sky scenario, based on simulations using modern climate

conditions from the year 1980. The calculated values for $\eta_C$ and $\eta_W$ are then used to calculate the corresponding optical depths $\lambda_C$ and $\lambda_W$ for the case of the modern atmosphere, and these are used to solve for $k_C$ and $k_W$, which then appear in the $G_C$ and $G_{W2}$ terms respectively, in Table 1 and Equation (7).

The vertical transport of sensible and latent heat is a difficult and complicated process to model, so many approximations are made to keep it within the scope of this work. For more details, see (Kypke (2019)). The heat transports are modelled via

*bulk aerodynamic exchange* formulae describing fluxes between the surface and the atmosphere as described in (Pierrehumbert




(2010); Hartmann (2016)). For the sensible heat flux, $c_p$ is the specific heat of the air being heated, and $T$ is its temperature. The bulk aerodynamic formula for sensible heat (SH) is

$$SH = c_p \, \rho \, C_{DS} \, U(T_S - T_A), \tag{20}$$

where $C_{DS}$ is the drag coefficient for temperature and $U$ is the mean horizontal wind velocity. The density of the atmosphere
$\rho$ is determined as a function of both surface temperature $T_S$ and altitude $Z$ using the barometric formula, and a constant lapse rate $\Gamma$ (ICAO (1993)) is used to determine the temperature gradient.

In the case of latent heat (LH), the moisture content is represented by $L_v r$, where $L_v$ is the latent heat of vaporization of water and $r$ is the *mass mixing ratio* of condensable air to dry air (Pierrehumbert (2010)),

$$LH = L_v \, \rho \, C_{DL} \, U(r_S - r_A), \tag{21}$$

where $C_{DL}$ is the drag coefficient for moisture. The mass mixing ratio is equal to the saturation mixing ratio times the relative humidity. The saturation mixing ratio depends on the saturation vapour pressure, which is a function of temperature as given by the Clausius-Clapeyron equation (6). The sensible and latent heat transports are combined into a single term, $F_C$, which replaces the $F_C$ term that was introduced in Figure 1 and Section 2. This term is defined here as a function of surface temperature $T_S$,

$$
\begin{aligned}
F_C ={}& c_p \, \frac{P_0}{R_A(T_S - \Gamma Z)} \left( \frac{T_S}{T_S - \Gamma Z} \right)^{-g/R_A \Gamma} C_{DS} \, U \, \Gamma Z \\
&+ \frac{L_v \, C_{DL} \, U}{R_A(T_S - \Gamma Z)} \left[ \varepsilon P^{sat}(T_R) \left( \frac{T_S}{T_S - \Gamma Z} \right)^{-g/R_A \Gamma} \exp \left( \frac{L_v}{R_W} \left[ \frac{1}{T_R} - \frac{1}{T_S} \right] \right)(1 - \delta) \right. \\
&+ \left. \delta \frac{L_v}{R_W(T_S - \Gamma Z)^2} \left( \varepsilon P^{sat}(T_R) \exp \left( \frac{L_v}{R_W} \left[ \frac{1}{T_R} - \frac{1}{T_S - \Gamma Z} \right] \right) \cdot \Gamma Z \right) \right] \\
={}& U \, \frac{P_0}{R_A(T_S - \Gamma Z)} \left( \frac{T_S}{T_S - \Gamma Z} \right)^{-g/R_A \Gamma} \\
&\left[ c_p \, C_{DS} \, \Gamma Z + L_v \, C_{DL} \, \varepsilon \cdot \left( \frac{P^{sat}(T_R)}{P_0} \exp \left( \frac{L_v}{R_W} \left[ \frac{1}{T_R} - \frac{1}{T_S} \right] \right) \right)(1 - \delta) \right. \\
&+ \left. \frac{\delta \, L_v \, \Gamma Z}{R_W(T_S - \Gamma Z)^2} \left( \frac{P^{sat}(T_R)}{P_0} \left( \frac{T_S}{T_S - \Gamma Z} \right)^{g/R_A \Gamma} \exp \left( \frac{L_v}{R_W} \left[ \frac{1}{T_R} - \frac{1}{T_S - \Gamma Z} \right] \right) \right) \right].
\end{aligned}
\tag{22}
$$

Here, $P_0$ is the atmospheric pressure at the surface $(Z = 0)$ and $R_A$ is the ideal gas constant specific to dry air. This equation is scaled by $\frac{1}{\sigma T_R^4}$ to nondimensionalize it, creating $f_C = \frac{F_C}{\sigma T_R^4}$ in (10), where $T_R = 273.15$ K is the reference temperature. As this $f_C$ represents energy moving from the surface to the atmosphere, it is subtracted from the surface equation (4) and added to the atmosphere equation (5). A different model of $F_C$, used by the authors in (Dortmans et al. (2019)), was a simple functional form calibrated to empirical data. The result was a relationship between $F_C$ and $T_S$ that is quantitatively very similar to that
given by (22). In (Kypke and Langford (2020)), $F_C$ was set equal to zero for the paleoclimate Arctic and Antarctic models for simplicity, since both *SH* and *LH* are very small for below freezing temperatures.





**Acknowledgements**

The authors gratefully acknowledge support of the Natural Sciences and Engineering Research Council of Canada and thank M. Garvie for generous assistance.

**Code/Data availability**

There are no supplemental Code or Data files.

**Author contributions**

The original conception of this work was due to W. Langford. Most of the mathematical analysis and computations were performed by K. Kypke, in his M.Sc.Thesis (Kypke (2019)). A. Willms co-supervised the work and provided valuable insights.

**Competing interests**

The authors declare no competing interests.



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



| Variables | Symbol | Value |
|---|---|---|
| Mean temperature of the surface | $T_S$ | -50 to + 40 C |
| Infrared radiation from the surface | $I_S = \sigma T_S^4$ | 141 to 419 W m$^{-2}$ |
| Mean temperature of the atmosphere | $T_A$ | -70 to + 10 C |
| Energy emitted by the atmosphere | $I_A = \sigma T_A^4$ | 87 to 219 W m$^{-2}$ |
| **Parameters and Constants** | **Symbol** | **Value** |
| Temperature of freezing point for water | $T_R$ | 273.15 K |
| Stefan-Boltzmann constant | $\sigma$ | $5.67 \times 10^{-8}$ W m$^{-2}$ K$^{-4}$ |
| Emissivity of dry air | $\varepsilon$ | 0.9 |
| Greenhouse gas absorptivity | $\eta$ | 0 to 1 |
| Absorptivity for $CO_2$ | $\eta_C$ | 0 to 1 |
| Absorptivity for $H_2O$ | $\eta_W$ | 0 to 1 |
| Absorptivity for clouds | $\eta_{Cl}$ | 0.255 |
| Portion of $I_A$ reaching surface | $\beta$ | 0.63 |
| Ocean heat transport | $F_O$ | 10 W m$^{-2}$ |
| Atmospheric heat transport | $F_A$ | 104 W m$^{-2}$ |
| Vertical heat conduction and latent heat | $F_C$ | 20 to 120 W m$^{-2}$ |
| Absorption of solar radiation | $F_S$ | $(1-\alpha)Q$ |
| Incident solar radiation at Poles | $Q_P$ | 173.2 W m$^{-2}$ |
| Incident solar radiation at Equator | $Q_E$ | 418.8 W m$^{-2}$ |
| Fraction of insolation absorbed by atmosphere | $\xi_A$ | 0.2324 |
| Fraction of insolation reflected by clouds | $\xi_R$ | 0.1212 (poles) 0.2235 (global) |
| Molar concentration of $CO_2$ in ppm | $\mu$ | 270 to 2000 ppm |
| Relative humidity of $H_2O$ | $\delta$ | 0 to 1 |
| Absorption coefficient for $CO_2$ | $k_C$ | 0.07424 m$^2$ kg$^{-1}$ |
| Absorption coefficient for $H_2O$ | $k_W$ | 0.05905 m$^2$ kg$^{-1}$ |
| Warm surface albedo for ocean | $\alpha_w$ | 0.08 |
| Cold surface albedo for Arctic | $\alpha_c$ | 0.7 |
| Albedo transition rate (in tanh function) | $\omega = \Omega/T_R$ | 0.01 |
| Standard atmosphere lapse rate | $\Gamma$ | $6.49 \times 10^{-3}$ K m$^{-1}$ |
| Normalized standard lapse rate | $\gamma = \Gamma/T_R$ | $2.38 \times 10^{-5}$ m$^{-1}$ |
| Tropopause height at North Pole | $Z_P$ | 9000 m |
| Latent heat of vaporization of water | $L_v$ | $2.2558 \times 10^6$ m$^2$ s$^{-2}$ |
| Universal ideal gas constant | $R$ | 8.3145 kg m$^2$ s$^{-2}$ K$^{-1}$ mol$^{-1}$ |
| Ideal gas constant specific to water vapour | $R_W$ | 461.4 m$^2$ s$^{-2}$ K$^{-1}$ |
| Saturated partial pressure of water at $T_R$ | $P_W^{sat}(T_R)$ | 611.2 Pa |
| Saturated density of water at $T_R$ | $\rho_W^{sat}(T_R)$ | $4.849 \times 10^3$ kg m$^{-3}$ |
| Greenhouse gas coefficient for $CO_2$ | $G_C$ | $1.162 \times 10^{-3}$ |
| Greenhouse gas coefficient 1 for $H_2O$ | $G_{W1}$ | 17.89 |
| Greenhouse gas coefficient 2 for $H_2O$ | $G_{W2}$ | 12.05 |
| Surface heat rate constant | $c_S$ | 6.53 W year m$^{-2}$ K$^{-1}$ |
| Atmosphere heat rate constant | $c_A$ | 0.1049 year |

**Table 1.** Summary of variables and parameters used in the EBM. The values of the physical constants $\xi_a, \xi_R, k_C, k_W, G_C, G_{W1}, G_{W2}$ are as determined in (Kypke (2019); Dortmans et al. (2019)).





| Year | Scenario | $F_A$ (W/m$^2$) | $F_O$ (W/m$^2$) |
|------|----------|-------------|-------------|
| 1850 | Historical | 107.28 | 9.75 |
| 2100 | RCP 2.6 | 104.03 | 13.00 |
| 2100 | RCP 8.5 | 97.53 | 19.50 |

**Table 2.** Atmosphere and ocean heat fluxes into the Arctic as simulated in (Koenigk and Brodeau (2014)), using the global coupled climate model EC-Earth.

| | $\tau_S$ | °C |
|---|------|-----|
| $\mu = 270$ | 1.0542 | 14.812 |
| $\mu = 540$ | 1.0709 | 19.363 |
| ECS | 0.016663 | 4.5516 |

**Table 3.** ECS (in °C) for the globally averaged EBM.

Arctic Region

| | $\tau_S$ | °C |
|---|------|-----|
| $\mu = 270$ | 0.8853 | -31.34 |
| $\mu = 540$ | 0.9145 | -23.34 |
| ECS | 0.0293 | 7.995 |

Antarctic Region

| | $\tau_S$ | °C |
|---|------|-----|
| $\mu = 270$ | 0.8671 | -36.30 |
| $\mu = 540$ | 0.8947 | -28.77 |
| ECS | 0.0276 | 7.531 |

Tropic Region

| | $\tau_S$ | °C |
|---|------|-----|
| $\mu = 270$ | 1.091 | 24.74 |
| $\mu = 540$ | 1.095 | 25.87 |
| ECS | 0.0041 | 1.128 |

**Table 4.** ECS values (in °C) for each of the three regional EBMs. The ECS is much greater for the Poles than for the Tropics, in agreement with observations.