# Peer review of "Anthropocene climate bifurcation"

_Nonlinear Processes in Geophysics, 2020_

## Referee Comment (RC1) · Peter Ditlevsen (Referee) · 10 Mar 2020

In this paper an energy balance model (EBM), presented in (Dortmans, 2019), containing a surface (represented by a surface temperature) and a single layer atmosphere (represented by the atmospheric radiation) is forced with increasing greenhouse gas concentrations following the four IPCC RCP scenarios. The EBM can represent different zonal bands, or the polar regions with oceanic and atmospheric heat fluxes across the zonal boundary. The model includes ice albedo – and water vapor feedbacks and an implicit ocean and atmosphere heat transport feedback.

The model has a bimodal regime with a saddle node bifurcation to a warm polar region state, which will be reached after the next century if the $CO_2$ level reaches 2000 ppm. The mechanism responsible for the bifurcation structure is via the sigmoidal dependence of ice albedo on temperature, essentially similar to Budyko-Sellers EBM. In

the parameter space spanned by the atmospheric CO2 concentration, the water vapor relative humidity, the oceanic heat transport and the steepness of the albedo switch, there is a cusp separating the fold and a mono-stable state.

The paper is well written, and I recommend publication. The authors might, however, consider a few revisions, which in my view would make it even more readable:

First of all, it is always a delicate balance, how much material to repeat from previous papers, in this case the reference above introducing the model. In my view it is either or: Either the reader is required to also consult other papers, or the paper should be fully self-contained. In the latter case a few additions would be helpful:

Explain the asymmetry between eqs. 1 and 2: Why use I_A and not T_A as variable? As is now, both have dimensions of W/m^2, with the consequence of different dimensionalities for c_S and c_A. This confused me at first.

Eq 10 for vertical heat transport f_C seems overly complicated for such a simple EBM. It is stated that the formula is obtained (derived, I take it) in (Kypke 2019. This a PhD thesis, which is not easily accessible for the reader. Consider at least hinting at where it comes from. Perhaps even a graph, f_C as a function of T_S. Though hard to read, I think there's a ")" missing (same goes for eq 22).

Now the mathematical analysis begins with Eqs 15 and 16: It would be helpful to remind the reader that mu enters via eta (through eq. 9), since mu will be the "control parameter" in the following.

A few more remarks:

Figure 6 (a) shows the responses in the four cases before the bifurcation point. These responses are quite linearly related to the RCPs (Figure 4). The same goes for the GCM scenarios presented in IPCC AR5 (Fig AI.8). The statement (line 239) that the EBM results are in good agreement with the GCM projections is thus an overstatement, both are related to the RCPs.
The bifurcation for the Arctic and not the global EBM depends critically on the oceanic heat transport F_0. It would be useful with a comparison of this EBM with the classical 1-d EBMs where meridional heat transport is modelled as a diffusion.

---

## Author Comment (AC1) · 18 Mar 2020

The authors wish to thank Referee #1 for a careful reading of the manuscript and for insightful comments.

The referee recognizes the delicate balance between making the paper self-contained, and requiring the reader to consult earlier work. The authors will improve the readability of the manuscript, with the following clarifications and additions, as suggested by the referee.

1. The confusing asymmetry between equations (1) and (2) will be explained, and the use of $I_A$ instead of $T_A$ will be better motivated.

2. Eq. (10) for the vertical heat transport $f_C$ will be simplified, and its derivation will be given in the Appendix.

3. The crucial dependence of eta on mu will be clarified in eqs. (15)(16).

4. The relationship between the EBM responses to the year 2100, shown in Fig. 6(a), to the GCM scenarios presented in IPCC AR5, Fig. AI.8, will be clarified. Both are largely determined by the RCPs.

5. The bifurcation for the Arctic EBM depends critically on the ocean heat transport $F_O$. The referee suggests that it would be useful to compare this EBM with classical 1D EBMs where meridional heat transport is modelled as a diffusion. The authors would be grateful if the referee could cite a paper (or papers) with which the meridional heat transport can be compared.

The authors assume that there will be a second referee report and will resubmit a revised manuscript after the second report.

---

## Referee Comment (RC2) · Michel Crucifix (Referee) · 4 May 2020

This article provides an improvement on the classical energy balance model (EBM) provided by the prognostic equations (1) and (2). These prognostic equations are the same as those nowadays found in textbooks, and based on the pioneering works of Budyko and Sellers. It is applied at a regional scale, hence a $F_o$ term accounts for the net supply of heat by ocean transport. Compared to the textbook formulations, the diagnostic equations are more elaborate. These are equation (6) to (10), and include specific parameterisations for sensible and latent heat transport, with details given in the lead author master's thesis. The albedo feedback given by equation (8) follows the fairly standard hyperbolic tangent formulation, and it is here presented as a consequence of the dynamics of sea ice.

Expectedly, the model presents bifurcations caused by the albedo dependence on temperature, as already studied by Budyko and Sellers, North, Ghil, and others. The claim of the authors is that the diagnostic equations have been given more attention in the present contribution, and are more accurate than in previous works, such that their model can be used for actual predictions of climate in the 21st century and beyond (most plots extend to 2300).

Specifically, the authors present previous EBMs as "lacking in the geophysical details and mathematical rigour required to make useful predictions". They go on: "This paper presents an EBM built upon basic laws of geophysics" and "it provides new mathematical evidence signifying that catastrophic climate change in polar regions is inevitable in the coming decades and centuries if current anthropogenic forcing continues unabated". This conclusion follows from a bifurcation analysis of the model, and the identification of co-dimension 2 (cusp) bifurcations in the parameter space spanned by the CO2 concentration and the net heat flux penetrating the domain studied.

Overall, I support the publication of this contribution. However, I believe that the positioning of this study with respect to the state-of-the-art is arguable. On the one hand, the claim that earlier studies have been "lacking mathematical rigour" seems somewhat excessive (the recent review of Ghil and Lucarini, 2019, arXiv 1910.00583, is a nice entry point, which shows how much mathematics has already gone in previous works). I would also not say that they did not provide "useful" predictions. On the other hand, they present general circulation models (GCM) as "too stable" to provide reliable warning on the sudden catastrophic events, citing Valdes (2011). Indeed, low-dimensional models tend to have a more clear-cut bifurcation structure than high-dimensional models such as GCMs, but there are also good reasons for the smoother character of certain transitions in the GCMs (spatial patterns, partial capture of scaling laws, effects of turbulence). In particular, a number of simulations with so-called earth models of intermediate complexity have been run well into the future and they do not necessarily present such "catastrophic" transitions; it would be legitimate to ask why they didn't so,

while they certainly include more "geophysical detail" than the current study. The interest of an EBM bifurcation analysis is not so much to provide an accurate prediction that would supersede the current state-of-the-art. It rather lies in sensitivity analysis and examination of the conditions that would generate a bifurcation (see, in particular, my comments below regarding line 109).

I would also like to make a few comments about the semantics.

There is an important distinction to be made between the existence of a bifurcation, and the potentially abrupt character of a transition (l. 50). The bifurcation, at least the way it is presented here, tells us about the topology of the attractor, which is a measure of the invariant manifold. The "abruptness" relates to the dynamics of the transient changes that occur when the system moves from one fixed point to the other. The existence of a bifurcation does not imply abruptness. Think of the melting of large ice sheets (incidentally, lacking in the present model). One way to address this ambiguity is to consider the dynamics of non-autonomous systems, as done for example with "mathematical rigour" by Ashwin et al. (2012, doi: 10.1098/rsta.2011.0306).

Furthermore, I would advise caution with the arguments that using "basic laws of geophysics" generates more accuracy (ll. 30-35). No climate model is computed "ab initio". Any model requires parameterisations which always include an empirical component, and the very simple EBM presented here is certainly no exception. Furthermore, the dynamics of climate are also related to biology and therefore involves knowledge and arguments that go beyond geophysics.

With these reservations expressed, I would like to reiterate that I am overall supportive to the publication of this paper, and we now proceed with the line-by-line comments.

1. l. 109 - The difference between $\alpha_c$ and $\alpha_w$ is quite large, and it seems pretty clear that the bifurcation depends on the amplitude of this difference, and of the slope of $\alpha(T)$ curve (as can be seen with a Lamerey diagram, in the way done by, e.g.,

Brovkin et al., 10.1029/1998JD200006). In the real world, the spatial distribution and seasonal cycles of snow and ice are likely to effectively smooth the albedo dependency expressed by equation (8).

2. l. 174 - $\beta_1$ and $\beta_2$: aren't these $\zeta_1$ and $\zeta_2$ ?

3. l. 176 - Start with a section 3.1.

4. l. 210 - Given that the definition of albedo function is so important for the existence (and significance) of bifurcations, it is necessary to be very explicit about the latitudes covered.

5. Figure 6 - Again, the arrow sketched on figure b should not be interpreted as if the transition was instantaneous. More generally, what are drawn here are steady states, while actual trajectories depart from the steady state.

6. Section 3.2.2 - Whether the heat flux will increase with global warming is arguable. For example, if Greenland melts, the thermohaline circulation may reduce in intensity, and so would the supply of heat to the high latitudes. If sea ice melts, the thermal contrast between the Atlantic and Arctic will also change, and the consequences on ocean and atmospheric transport are not necessarily trivial.

7. l. 304 - "the EBM predicts that CO2 mitigation strategies, if introduced soon enough, may avert the drastic consequences of this bifurcation." We have to be careful about writing such sentences. This paper is not about mitigation strategies. We are not discussing what would be the effect of policies on CO2 emissions and CO2 concentrations. So the EBM predicts nothing about mitigation strategies. It only predicts that conservative RCP scenarios avoid the bifurcation.

8. l. 319 - It might be good to write somewhere explicitly that land ice masses are considered as constant in this study (I might have missed this, though)

9. l. 382 : the IPCC is "reporting" (results of published work), not "predicting".

10. l. 398 - It is correct that for 4.55° C is a reasonable number, though on the high range, but the last part of the argument seems a little bit overstretched. Some GCMs have a low climate sensitivity, too, and yet they rely on what the authors call "geophysics".

11. l. 400 : "based on geophysics rather than statistical data" : I would write "based on physical rather than statistical modelling" (though, to be fair, several assumptions included in the physical model are based on statistical modelling or regressions, but we understand what is being said here).

12. l. 409 - "the analysis of this paper presents a mathematical proof that a bifurcation can occur in an EBM" : this is correct but again I would do justice to other authors who already presented bifurcation analysis in EBMs.

13. l.423 - The author nicely present their plans for the next years with the project to develop a 3D model. What is the inteded added value compared to existing initiatives like PLASIM ?

14. Code availability: Especially given the stance on open source in the conclusion paragraph, I would strongly encourage the authors to provide the code necessary to reproduce the main results, either in the form of a version-controlled repository (e.g.: gitlab), or an doi-ed archive (e.g. zenodo).

---

## Author Comment (AC2) · 7 May 2020

The authors thank Referee #2 for his helpful and constructive criticisms of the manuscript.

The referee states that this contribution needs to be better positioned with respect to the state-of-the-art. The paper appears to downplay important contributions of previous researchers. This was not intentional. The referee quotes several sentences from the Introduction to make this point. The authors will rewrite those introductory paragraphs, to better position this contribution in the discipline. The referee then points out that the interest of an EBM bifurcation analysis is not to provide an accurate prediction that would supersede the current state-of-the-art. It can provide a closer examination of the conditions that would generate a bifurcation.

The referee correctly points out that the existence of a bifurcation does not neces-

sarily imply an abrupt transition. However, the particular bifurcations found in this manuscript are organized around a codimension-two cusp bifurcation, which yields the phenomenon of hysteresis, and that does imply the possibility of an abrupt transition to a new equilibrium. After the mutual annihilation of the saddle and the node in a saddlenode bifurcation, trajectories pass slowly through the so-called "ghost equilibrium" in a neighbourhood where the saddlenode had been, see item 5. below. Outside of that neighbourhood, the time of the transition to the new equilibrium state is determined by the inverses of the rate constants $c_S$ and $c_A$ in equations (1) and (2), which are of order one (not large).

Also, the referee is correct in saying that the very simple EBM presented here includes an empirical component, and in fact does not rest exclusively on the basic laws of geophysics. The authors will clarify this in the manuscript.

Line-by-line comments:

1. Yes, the difference between $\alpha_c$ and $\alpha_w$ is quite large, but these values are taken from the literature. The smoothing effect of spatial distribution and seasonal cycles have been taken into account in equation (8).

2. Yes, $\beta_1$ and $\beta_2$ should be $\zeta_1$ and $\zeta_2$.

3. The authors feel that the section numbers are fine as they are.

4. The latitudes included in the Arctic model (and Antarctic) will be made explicit, in the final submission.

5. The authors agree that the transition indicated by an arrow in Figure 6 should not be interpreted as instantaneous. In fact, just to the right of the saddlenode bifurcation point, trajectories in the phase space move upward slowly through a neighbourhood that is sometimes called the "ghost" of the saddlenode (with transit time that has inverse square-root dependence on the unfolding parameter in that neighbourhood). Outside of that neighbourhood, trajectories have their normal velocity. The captions of Figure

6-11 will be rewritten accordingly.

6. The expected changes of heat flux, due to ocean and atmospheric transport, are matters of debate today. Some of this debate has been reflected in the manuscript. The melting of the Greenland ice cap surely will affect the North Atlantic - Arctic thermohaline circulation, in some way. The authors are not qualified to partake of this debate; however, this model is easily adapted to any heat flux scenario that is proposed.

7. Of course, the referee is right that this paper is not about mitigation strategies, only about the effects of various rates of $CO_2$ emissions. The IPCC is concerned about mitigation strategies and the effects of policy decisions. The Representative Concentration Pathways (RCP) are mechanisms that were invented to simulate the possible effects of different mitigation strategies. This paragraph will be rewritten.

8. Correct, this study does not consider the loss of land ice masses, other than, as stated, the loss of polar ice-caps will cause sea levels to rise, possibly increasing the rate of ocean heat transport.

9. Point taken.

10. We are happy with our Equilibrium Climate Sensitivity (ECS) value of 4.55 C calculated for the global EBM, which is at the high end of the IPCC range. It was [Priostosescu and Huybers (2017)] who explained that statistical models tend to lie at the low end of the IPCC range, while deterministic nonlinear models like ours are at the high end. We can not explain why some GCM's have a lower climate sensitivity without examining them individually.

11. Yes, we agree, the word "geophysics" is used inappropriately here (and also in point 10). What was meant was that deterministic, nonlinear physical reasoning is at the foundation of the model, not statistical modelling.

12. The referee is right; there have been plenty of EBM's that present bifurcations, going back to Budyko, Sellers, North and others. However we have, I believe, given

the first mathematical proof of the existence of a cusp bifurcation in an EBM, complete with the determination of the corresponding Center Manifold and the physical unfolding parameters. The complete mathematical analysis is given in a different paper of the authors [Kypke and Langford (2020)], for a slightly different EBM, that is our paleoclimate model. The present paper extends that analysis to the Anthropocene.

13. The model presented in this paper is just a first step in a program of research that eventually will combine the surface + atmosphere energy balance ideas of this model with the convective flow of the Navier-Stokes-Boussinesq spherical shell model in [Lewis and Langford (2008)]. Greg Lewis still has the code for that convection model, so it will be relatively easy to add the energy balance as in this model, to determine meridional surface temperatures implicitly. Greg has already proven the existence of a cusp bifurcation in that PDE model. Another goal is to adapt an open source model of intermediate complexity, such as perhaps PLASIM, to finding bistability and bifurcations, and then to compare results.

14. Code availability: We are currently investigating our options in providing the code for our results in an online repository.

Papers cited by the referee will be added as references in the manuscript.

---

## Author Response (AR1)

**npg-2020-4 : Anthropocene climate bifurcation**
**AUTHORS' RESPONSE**

**Kolja Leon Kypke, William Finlay Langford, and Allan Richard Willms**

Department of Mathematics and Statistics, University of Guelph, 50 Stone Road West, Guelph, ON, Canada N1G 2W1

**Correspondence:** William F. Langford (wlangfor@uoguelph.ca)

**1 Comments from referees**

**1.1 Referee 1 comments**

Peter Ditlevsen (Referee) pditlev@nbi.ku.dk

In this paper an energy balance model (EBM), presented in (Dortmans, 2019), containing a surface (represented by a surface temperature) and a single layer atmosphere (represented by the atmospheric radiation) is forced with increasing greenhouse gas concentrations following the four IPCC RCP scenarios. The EBM can represent different zonal bands, or the polar regions with oceanic and atmospheric heat fluxes across the zonal boundary. The model includes ice albedo and water vapor feedbacks and an implicit ocean and atmosphere heat transport feedback.

The model has a bimodal regime with a saddle node bifurcation to a warm polar region state, which will be reached after the next century if the $CO_2$ level reaches 2000 ppm. The mechanism responsible for the bifurcation structure is via the sigmoidal dependence of ice albedo on temperature, essentially similar to Budyko-Sellers EBM. In the parameter space spanned by the atmospheric $CO_2$ concentration, the water vapor relative humidity, the oceanic heat transport and the steepness of the albedo switch, there is a cusp separating the fold and a mono-stable state. The paper is well written, and I recommend publication. The authors might, however, consider a few revisions, which in my view would make it even more readable:

First of all, it is always a delicate balance, how much material to repeat from previous papers, in this case the reference above introducing the model. In my view it is either or: Either the reader is required to also consult other papers, or the paper should be fully self-contained. In the latter case a few additions would be helpful:

Explain the asymmetry between eqs. 1 and 2: Why use $I_A$ and not $T_A$ as variable? As is now, both have dimensions of $Wm^{-2}$, with the consequence of different dimensionalities for $c_S$ and $c_A$. This confused me at first.

Eq 10 for vertical heat transport $f_C$ seems overly complicated for such a simple EBM. It is stated that the formula is obtained (derived, I take it) in (Kypke 2019). This a PhD thesis, which is not easily accessible for the reader. Consider at least hinting at where it comes from. Perhaps even a graph, $f_C$ as a function of $T_S$. Though hard to read, I think there is a ")" missing (same goes for eq 22).

Now the mathematical analysis begins with Eqs 15 and 16: It would be helpful to remind the reader that $\mu$ enters via $\eta$ (through eq. 9), since $\mu$ will be the "control parameter" in the following.

A few more remarks:

Figure 6 (a) shows the responses in the four cases before the bifurcation point. These responses are quite linearly related to the RCPs (Figure 4). The same goes for the GCM scenarios presented in IPCC AR5 (Fig AI.8). The statement (line 239) that the EBM results are in good agreement with the GCM projections is thus an overstatement, both are related to the RCPs.

The bifurcation for the Arctic and not the global EBM depends critically on the oceanic heat transport $F_O$. It would be useful with a comparison of this EBM with the classical 1-d EBMs where meridional heat transport is modelled as a diffusion.

**1.2 Referee 2 comments**

Michel Crucifix (Referee) michel.crucifix@uclouvain.be

This article provides an improvement on the classical energy balance model (EBM) provided by the prognostic equations (1) and (2). These prognostic equations are the same as those

nowadays found in textbooks, and based on the pioneering works of Budyko and Sellers. It is applied at a regional scale, hence a $F_O$ term accounts for the net supply of heat by ocean transport. Compared to the textbook formulations, the diagnostic equations are more elaborate. These are equation (6) to (10), and include specific parametrisations for sensible and latent heat transport, with details given in the lead author master?s thesis. The albedo feedback given by equation (8) follows the fairly standard hyperbolic tangent formulation, and it is here presented as a consequence of the dynamics of sea ice.

Expectedly, the model presents bifurcations caused by the albedo dependence on temperature, as already studied by Budyko and Sellers, North, Ghil, and others. The claim of the authors is that the diagnostic equations have been given more attention in the present contribution, and are more accurate than in previous works, such that their model can be used for actual predictions of climate in the 21st century and beyond (most plots extend to 2300). Specifically, the authors present previous EBMs as "lacking in the geophysical details and mathematical rigour required to make useful predictions". They go on: "This paper presents an EBM built upon basic laws of geophysics" and "it provides new mathematical evidence signifying that catastrophic climate change in polar regions is inevitable in the coming decades and centuries if current anthropogenic forcing continues unabated". This conclusion follows from a bifurcation analysis of the model, and the identification of co-dimension 2 (cusp) bifurcations in the parameter space spanned by the $CO_2$ concentration and the net heat flux penetrating the domain studied.

Overall, I support the publication of this contribution. However, I believe that the positioning of this study with respect to the state-of-the-art is arguable. On the one hand, the claim that earlier studies have been "lacking mathematical rigour" seems somewhat excessive (the recent review of Ghil and Lucarini, 2019, arXiv 1910.00583, is a nice entry point, which shows how much mathematics has already gone in previous works). I would also not say that they did not provide "useful" predictions. On the other hand, they present general circulation models (GCM) as "too stable" to provide reliable warning on the sudden catastrophic events, citing Valdes (2011). Indeed, low-dimensional models tend to have a more clear-cut bifurcation structure than high-dimensional models such as GCMs, but there are also good reasons for the smoother character of certain transitions in the GCMs (spatial patterns, partial capture of scaling laws, effects of turbulence). In particular, a number of simulations with so-called earth models of intermediate complexity have been run well into the future and they do not necessarily present such "catastrophic" transitions; it would be legitimate to ask why they didn't so, while they certainly include more "geophysical detail" than the current study. The interest of an EBM bifurcation analysis is not so much to provide an accurate prediction that would supersede the current state-of-the-art. It rather lies in sensitivity analysis and examination

of the conditions that would generate a bifurcation (see, in particular, my comments below regarding line 109). I would also like to make a few comments about the semantics.

There is an important distinction to be made between the existence of a bifurcation, and the potentially abrupt character of a transition (l. 50). The bifurcation, at least the way it is presented here, tells us about the topology of the attractor, which is a measure of the invariant manifold. The "abruptness" relates to the dynamics of the transient changes that occur when the system moves from one fixed point to the other. The existence of a bifurcation does not imply abruptness. Think of the melting of large ice sheets (incidentally, lacking in the present model). One way to address this ambiguity is to consider the dynamics of non-autonomous systems, as done for example with "mathematical rigour" by Ashwin et al. (2012, doi: 10.1098/rsta.2011.0306). Furthermore, I would advise caution with the arguments that using "basic laws of geophysics" generates more accuracy (ll. 30-35). No climate model is computed "ab initio". Any model requires parametrizations which always include an empirical component, and the very simple EBM presented here is certainly no exception. Furthermore, the dynamics of climate are also related to biology and therefore involves knowledge and arguments that go beyond geophysics. With these reservations expressed, I would like to reiterate that I am overall supportive to the publication of this paper, and we now proceed with the line-by-line comments.

1. l. 109 - The difference between $\alpha_c$ and $\alpha_w$ is quite large, and it seems pretty clear that the bifurcation depends on the amplitude of this difference, and on the slope of $\alpha(T)$ curve (as can be seen with a Lamerey diagram, in the way done by, e.g., Brovkin et al., 10.1029/1998JD200006). In the real world, the spatial distribution and seasonal cycles of snow and ice are likely to effectively smooth the albedo dependency expressed by equation (8).

2. l. 174 - $\beta_1$ and $\beta_2$: aren't these $\zeta_1$ and $\zeta_2$ ?

3. l. 176 - Start with a section 3.1.

4. l. 210 - Given that the definition of albedo function is so important for the existence (and significance) of bifurcations, it is necessary to be very explicit about the latitudes covered.

5. Figure 6 - Again, the arrow sketched on figure b should not be interpreted as if the transition was instantaneous. More generally, what are drawn here are steady states, while actual trajectories depart from the steady state.

6. Section 3.2.2 - Whether the heat flux will increase with global warming is arguable. For example, if Greenland melts, the thermohaline circulation may reduce in intensity, and so would the supply of heat to the high latitudes. If sea ice melts, the thermal contrast between the Atlantic and Arctic will also change, and the consequences on ocean and atmospheric transport are not necessarily trivial.

7. l. 304 - "the EBM predicts that $CO_2$ mitigation strategies, if introduced soon enough, may avert the drastic consequences of this bifurcation." We have to be careful about writing such sentences. This paper is not about mitigation

strategies. We are not discussing what would be the effect of policies on CO2 emissions and CO2 concentrations. So the EBM predicts nothing about mitigation strategies. It only predicts that conservative RCP scenarios avoid the bifurcation.

8. l. 319 - It might be good to write somewhere explicitly that land ice masses are considered as constant in this study (I might have missed this, though)

9. l. 382 : the IPCC is "reporting" (results of published work), not "predicting".

10. l. 398 - It is correct that for 4.55° C is a reasonable number, though on the high range, but the last part of the argument seems a little bit overstretched. Some GCMs have a low climate sensitivity, too, and yet they rely on what the authors call "geophysics".

11. l. 400 : "based on geophysics rather than statistical data": I would write "based on physical rather than statistical modelling" (though, to be fair, several assumptions included in the physical model are based on statistical modelling or regressions, but we understand what is being said here).

12. l. 409 - "the analysis of this paper presents a mathematical proof that a bifurcation can occur in an EBM": this is correct but again I would do justice to other authors who already presented bifurcation analysis in EBMs.

13. l.423 - The author nicely present their plans for the next years with the project to develop a 3D model. What is the intended added value compared to existing initiatives like PLASIM?

14. Code availability: Especially given the stance on open source in the conclusion paragraph, I would strongly encourage the authors to provide the code necessary to reproduce the main results, either in the form of a version-controlled repository (e.g.: gitlab), or an doi-ed archive (e.g. zenodo).

**2 Authors' response**

**2.1 Response to referee 1**

The authors wish to thank Referee 1 for a careful reading of the manuscript and for insightful comments.

The referee recognizes the delicate balance between making the paper self-contained, and requiring the reader to consult earlier work. The authors will improve the readability of the manuscript, with the following clarifications and additions, as suggested by the referee.

1. The confusing asymmetry between equations (1) and (2) will be explained, and the use of $I_A$ instead of $T_A$ will be motivated.

2. Eq. (10) for the vertical heat transport $f_C$ will be simplified, and its derivation will be given in the Appendix. The referee is right that there is a missing ")".

3. The crucial dependence of $\eta$ on $\mu$ and $\delta$ will be clarified in eqs. (15)(16).

4. The relationship between the EBM responses shown in Fig. 6(a), and the GCM scenarios presented in IPCC AR5 Fig. AI.8 and AI.9, up to the year 2100, will be clarified. Both use the RCP scenarios to determine hypothetical $CO_2$ concentrations ($\mu$) by year, out to year 2100. Then both use these $CO_2$ concentrations as input to the model (EBM or GCM) to determine predicted climate changes for the same period.

5. The bifurcation for the Arctic EBM depends critically on the ocean heat transport $F_O$. The referee suggests that it would be useful to compare this EBM with classical 1D EBMs where meridional heat transport is modelled as a diffusion. The authors' plan of future work will include determination of the meridional heat transport in more sophisticated models, and those results will be compared with this simple EBM presented here.

**2.2 Response to referee 2**

The authors thank Referee 2 for his constructive criticisms of the manuscript.

The referee states that this contribution needs to be better positioned with respect to the state-of-the-art. The paper appears to downplay important contributions of previous researchers. This was not intentional. The referee quotes several sentences from the Introduction to make his point. The authors will rewrite those introductory paragraphs, to better position this contribution in the discipline. The referee then correctly points out that the interest of an EBM bifurcation analysis is not to provide an accurate prediction that would supersede the current state-of-the-art. It can provide a closer examination of the conditions that would generate a bifurcation.

The referee points out that the existence of a bifurcation does not necessarily imply an abrupt transition. However, the particular bifurcations found in this manuscript combine to give the phenomenon of hysteresis, which does imply abrupt transition in a dynamical system. The authors will clarify this distinction. Also, the referee is correct in saying that the very simple EBM presented here includes an empirical component, and in fact does not rest exclusively on the basic laws of geophysics.

Line-by-line comments:

1. Yes, the difference between $\alpha_c$ and $\alpha_w$ is quite large, but these values are taken from the literature. The smoothing effect of spatial distribution and seasonal cycles have been taken into account in equation (8).

2. Yes, $\beta_1$ and $\beta_2$ should be $\zeta_1$ and $\zeta_2$. Thank you.

3. The authors feel that the section numbers are fine as they are.

4. The latitudes included in the Arctic model are made explicit in subsection 3.2.2. The Antarctic is similar.

5. The authors agree that the transition indicated by an arrow in Figure 6 should not be interpreted as instantaneous. In fact, just to the right of the saddlenode bifurcation point, trajectories in the phase space move upward very slowly,

through what is sometimes called the "ghost" of the saddlenode (the transit time has inverse square-root dependence on the unfolding parameter, in a small neighbourhood). Outside of that neighbourhood, trajectories have their normal velocity determined by $c_S$ and $c_A$. The captions and text for Figure 6–11 will be rewritten accordingly.

6. The changes of heat flux, due to ocean and atmospheric transport, are matters of debate today. Some of this debate has been reflected in the manuscript. The melting of the Greenland ice cap surely will affect the North Atlantic – Arctic thermohaline circulation, in some way. The authors are not qualified to partake of this debate; however, this model is easily adapted to any heat flux scenario that is proposed.

7. Of course, the referee is right that this paper is not about mitigation strategies, only about the effects of various rates of $CO_2$ emissions. The IPCC is concerned about mitigation strategies and the effects of policy decisions. The Representative Climate Pathways (RCP) are mechanisms invented to simulate the possible effects of different mitigation strategies. This paragraph will be rewritten as suggested by the referee.

8. Correct, this study does not consider the loss of land ice masses.

9. Point taken. The IPCC is reporting, not predicting. This will be changed.

10. We are happy with our Equilibrium Climate Sensitivity (ECS) value of 4.55 C calculated for the global EBM, which is at the high end of the IPCC range. It was Priostosescu and Huybers (2017) who pointed out that statistical models tend to lie at the low end of the IPCC range, while deterministic nonlinear models like ours are at the high end. We can not explain why some GCM's have a lower climate sensitivity without examining them individually.

11. Yes, we agree, the word "geophysics" is used inappropriately here (and also in point 10). What was meant was that deterministic, nonlinear physical reasoning is at the foundation of the model, not statistical modelling.

12. The referee is right; there have been plenty of EBM's that present bifurcations, going back to Budyko, Sellers, North and others. However we have, I believe, given the first mathematical proof of the existence of a cusp bifurcation in an EBM, complete with the determination of the corresponding Center Manifold and the physical unfolding parameters. The complete analysis is given in another paper of the authors (Kypke and Langford 2020), for a slightly different EBM, that is our paleoclimate model. The present paper extends that analysis to the Anthropocene.

13. The model presented in this paper is just a first step in a program of research that eventually will combine the surface – atmosphere energy balance assumptions of this model with the convective flow of the Navier-Stokes-Boussinesq spherical shell model as in [Lewis and Langford (2008)]. Greg Lewis still has the code for that convection model, so it will be relatively easy to add energy balance from this model, to determine meridional surface temperatures implicitly. Greg

has already proven the existence of a cusp bifurcation in that PDE model. Another goal is to adapt an open source model, such as perhaps PLASIM to our goals, in particular the goal of finding bistability and bifurcations, and then to compare the results.

14. Code availability: We are currently investigating our options in providing the code for our results in an online repository.

**3    Author's changes in manuscript**

The authors are mathematicians, and are well aware of the fact that they are novices in the field of climate science. The care and time given by the two referees, to better position this work in the climate change literature, is very much appreciated by the authors. The referees' suggestions in this regard have all been incorporated in the manuscript, in the manner indicated point-by-point in Section 2, above.

The confusing asymmetry, using $T_S$ in Eq. (1) but $I_A$ in Eq. (2), has been explained in the text, in the paragraph following Eq.s (1) and (2), and further explained in the derivation of Eq. (6). By "freeing" $T_A$ to vary with altitude, the authors were enabled to use the atmospheric temperature dependence with height as given by the *lapse rate* of the ICAO International Standard Atmosphere. Since $H_2O$ concentration varies with temperature, this allows a much more accurate determination of the greenhouse effect of water vapour. This choice is a significant improvement of the present EBM over previous EBMs in the literature.

Equations (10) and (22) for $f_C$ and $F_C$ have been simplified, as requested by Referee 1. Since $F_C$ is only a minor contributor to the energy balance, a high degree of precision in its formulation is not necessary.

The change in the formula for $F_C$ resulted in minor changes in some of the Figures. These have been recomputed. All figures have been configured to follow Copernicus guidelines.

The relation between Fig. 6(a) and Fig.s AI.8 and AI.9 in IPCCC AR4 has been clarified. Both use the four RCPs of Fig. 4 to determine hypothetical $CO_2$ concentrations ($\mu$) by year, out to year 2100. Then both use these $CO_2$ concentrations as input to the model (EBM or GCM) to determine predicted climate changes. The agreement between EBM and GCM predictions is good. This agreement lends confidence to the use of this EBM for climate change predictions to the year 2300.

Referee 2 highlights several sentences in the manuscript that appear to disparage previous contributions to the field, saying these were "lacking in mathematical rigour" or did not provide "useful" contributions. These statements have been removed in the revised manuscript. Instead, new citations have been added, which exhibit both rigour and useful contributions.

The meaning of the vertical arrows, indicating transitions in Figures 6, 7 and 10, has been better explained in the Figure captions and the text.

We have designated the results quoted from the IPCC AR4
5 as "reports" rather than "predictions", as pointed out by Referee 2.

All of the papers cited by the referees have been added to the bibliography of the paper, along with statements in the paper which position them relative to the present work. Other
10 relevant citations have been added as well, for a bibliography that now has 20 additional references, out of a total of 75.

In order to accommodate the two-column format, the figure pairs have been repositioned vertically instead of horizontally.

15 Following the Copernicus style instructions, we have used abbreviations Eq. and Fig. for equations and figures.

The "Conclusions" section has been expanded to better describe future work. The "Author Contribution" and "Competing Interests" sections have been completed.

20 The computer code used to produce the Figures of this manuscript has been made available on the zenodo website, as cited in the manuscript.

A complete listing of changes in the manuscript, generated by the software latexdiff, follows this Authors' Response.

[revised manuscript text omitted]